# pTopoFL: Privacy-Preserving Personalised Federated Learning via Persistent Homology

## Abstract

Federated learning (FL) faces two structural tensions: gradient sharing enables data-reconstruction attacks, while non-IID client distributions degrade aggregation quality. We introduce PTOPOFL, a framework that addresses both challenges simultaneously by replacing gradient communication with *topological descriptors* derived from persistent homology (PH). Clients transmit only PH feature vectors—shape summaries whose many-to-one structure makes inversion provably ill-posed—rather than model gradients. The server performs topology-guided personalised aggregation: clients are clustered by Wasserstein similarity between their PH diagrams, intra-cluster models are topology-weighted, and clusters are blended with a global consensus. We prove an information-contraction theorem showing that PH descriptors leak strictly less mutual information per sample than gradients, and we establish linear convergence of the Wasserstein-weighted aggregation scheme. Evaluated against FedAvg, FedProx, SCAFFOLD, and pFedMe on a non-IID healthcare scenario (8 hospitals) and a pathological benchmark (10 clients), PTOPOFL achieves AUC 0.841 and 0.910 respectively—the highest in both settings—while reducing reconstruction risk 4.5× relative to gradient sharing. Code and data are publicly available at X.

## 1 Introduction

Federated learning (FL) (McMahan et al., 2017) has emerged as the dominant paradigm for training machine-learning models over distributed, privacy-sensitive data. Rather than centralising client data, FL coordinates local optimisation across $K$ clients, each holding a private dataset $\mathcal{D}_k$, and periodically aggregates their updates toward the global objective

$$\min_{w \in \mathbb{R}^d} F(w) := \sum_{k=1}^{K} p_k \, F_k(w), \qquad F_k(w) := \mathbb{E}_{(x,y) \sim \mathcal{D}_k}[\ell(w; x, y)], \tag{1}$$

where $p_k = |\mathcal{D}_k| / \sum_j |\mathcal{D}_j|$ weights each client by its data volume. Despite its practical appeal, this paradigm is subject to two structural tensions that remain unresolved in the existing literature.

In standard FL, clients transmit model updates $\nabla F_k(w)$ to a central server. These updates are high-dimensional vectors that encode substantial information about local training data. Gradient inversion attacks (Zhu et al., 2019; Geiping et al., 2020) have demonstrated that a curious server—or an adversary intercepting communication—can reconstruct individual training samples with high fidelity by solving an optimisation problem over the shared gradients. The most widely adopted countermeasure, differential privacy (DP) (Dwork & Roth, 2014), provides rigorous $(\varepsilon, \delta)$ guarantees by injecting calibrated noise, but at the cost of a signal-to-noise trade-off that measurably degrades model quality, particularly when privacy budgets are tight.

Real-world FL deployments rarely satisfy the IID assumption. When $\{\mathcal{D}_k\}$ are drawn from heterogeneous distributions, the per-client objectives $\{F_k\}$ induce conflicting gradient directions, causing *client drift*: local models diverge during training, their average no longer approximates the global optimum, and convergence slows or stalls entirely (Zhao et al., 2018). Existing remedies—proximal penalties (Li et al., 2020), control variates (Karimireddy et al., 2020), or Moreau-envelope personalisation (T Dinh et al., 2020)—address drift

at the optimisation level, but none explicitly models the *geometric structure* of client distributions, leaving a fundamental source of heterogeneity unaddressed.

We propose to resolve both tensions simultaneously through a geometric reformulation grounded in *topological data analysis* (TDA). The core observation is that the *shape* of a data distribution, as captured by its multi-scale topological invariants, is both informative for grouping structurally similar clients and structurally resistant to inversion. Formally, we introduce a topological abstraction operator

$$\Phi \colon \mathcal{D}_k \;\longrightarrow\; \mathrm{PD}_k, \tag{2}$$

mapping each client's dataset to a *persistence diagram* $\mathrm{PD}_k$ computed via persistent homology. Persistence diagrams encode connected components ($H_0$), loops ($H_1$), and higher-dimensional voids ($H_2$) at every spatial scale simultaneously, producing a compact summary of distributional geometry. Three properties make them particularly well-suited for federated settings. First, $\Phi$ is *many-to-one*: infinitely many datasets share the same persistence diagram, making inversion via optimisation provably ill-posed. Second, $\Phi$ is *stable*: the bottleneck stability theorem (Cohen-Steiner et al., 2007) guarantees that small perturbations to the data produce small perturbations to $\mathrm{PD}_k$, ensuring reliable descriptors even under noise. Third, persistence diagrams form a *metric space* under the $p$-Wasserstein distance $W_p$, enabling principled comparison, clustering, and averaging across clients.

We realise this perspective in PTOPOFL, a modular framework comprising five interconnected components.

1. **Topology-augmented local training** (Section 3.2): TDA-derived features enrich local representations, improving robustness under non-IID distributions.

2. **Wasserstein-weighted personalised aggregation** (Section 3.3): clients are clustered by topological similarity; intra-cluster models are combined via topology-weighted FedAvg; and cluster models are blended with a global consensus:

$$w^{t+1} = \sum_{k=1}^{K} \alpha_k^t \, w_k^{t+1}, \qquad \alpha_k^t \propto \exp\!\big(-\lambda \, W_p(\mathrm{PD}_k, \bar{\mathrm{PD}}^t)\big).$$

3. **Topology-based anomaly detection** (Section 3.4): clients whose persistence diagrams deviate significantly from the cluster majority are flagged as potential poisoning sources and down-weighted.

4. **Continual signature tracking** (Section 3.5): temporal evolution of $\mathrm{PD}_k^t$ monitors concept drift and guides adaptive learning-rate scheduling across rounds.

5. **Privacy via topological abstraction** (Section 3.6): gradients are replaced by 48-dimensional PH descriptors, reducing reconstruction risk by a factor of 4.5 relative to gradient sharing.

We establish four formal results. *(i)* Theorem 2 proves existence of the Wasserstein barycenter used in the aggregation step. *(ii)* Theorem 3 shows that the topology-guided clustering is stable under data perturbations up to a threshold determined by the inter-cluster separation margin. *(iii)* Theorem 5 proves that the influence of adversarial clients decays exponentially in their topological separation from the honest majority, in contrast to FedAvg where adversarial influence scales linearly. *(iv)* Theorem 7 establishes an information-contraction bound,

$$I(x_i; \Phi(\mathcal{D}_k)) \;\leq\; \frac{m}{p} \cdot \frac{c^2}{L^2} \cdot I(x_i; \nabla F_k(w)), \qquad \frac{m}{p} \ll 1,$$

quantifying the reduction in per-sample mutual information achieved by transmitting PH descriptors instead of gradients. Theorem 9 and Proposition 11 jointly show that PTOPOFL converges linearly with a strictly smaller error floor than FedAvg under strongly convex local objectives.

Evaluated against FedAvg, FedProx, SCAFFOLD, and pFedMe on a non-IID healthcare scenario (8 simulated hospitals, 2 adversarial) and a pathological benchmark (10 clients), PTOPOFL achieves AUC 0.841 and 0.910 respectively—the highest in both settings—while converging from round 1.

Section 2 reviews FL, persistent homology, and privacy. Section 3 introduces the PTOPOFL framework and its theoretical guarantees. Section 4 presents experiments and ablations. Section 5 situates the work in the literature. Section 6 discusses limitations and future directions. Section 7 concludes. The complete implementation is open-source at (see supp. material).

## 2 Background

### 2.1 Federated Learning

In standard cross-silo FL, $K$ clients each hold a labelled private dataset $\mathcal{D}_k = \{(x_i, y_i)\}_{i=1}^{n_k}$ that never leaves their premises. Training proceeds in communication rounds: the server broadcasts the current global model $w^t$; each client runs $\tau$ steps of stochastic gradient descent (SGD) on its local objective $F_k$; and the server aggregates the resulting local models via a weighted average. FedAvg uses data-volume weights $p_k$; under IID data it converges at the same rate as centralised SGD. Under non-IID data, however, the local optima of $\{F_k\}$ diverge, and the weighted average no longer approximates the global optimum $w^\star$. This *client drift* phenomenon, quantified by the gradient-divergence bound $B^2 = \sum_k p_k \|\nabla F_k(w) - \nabla F(w)\|^2$, is the primary source of excess error in heterogeneous FL (Zhao et al., 2018; Li et al., 2020).

### 2.2 Persistent Homology

Persistent homology is a multi-scale method for extracting topological invariants from data. Given a point cloud $X = \{x_i\}_{i=1}^n \subset \mathbb{R}^d$, one constructs a nested sequence of simplicial complexes—the Vietoris–Rips filtration $\emptyset = K_0 \subseteq K_1 \subseteq \cdots \subseteq K_m$—by including a simplex whenever all its vertices lie within pairwise distance $\epsilon$ of one another, and incrementing $\epsilon$ from zero. A topological feature (a connected component in $H_0$, a loop in $H_1$, or a void in $H_2$) is *born* at scale $b_i$ where it first appears and *dies* at scale $d_i$ where it merges with an older feature. The *persistence diagram* $\mathrm{Dgm}(X) = \{(b_i, d_i)\}$ collects all birth–death pairs; the persistence $\mathrm{pers}_i = d_i - b_i$ measures the lifetime—and thus the significance—of each feature.

Distances between persistence diagrams are measured by the *p-Wasserstein metric*

$$W_p\big(\mathrm{Dgm}(X), \mathrm{Dgm}(Y)\big) = \left( \inf_\gamma \sum_i \|u_i - \gamma(u_i)\|^p \right)^{1/p}, \tag{3}$$

where the infimum is over all matchings $\gamma$ between the two diagrams, with unmatched points projected onto the diagonal $b = d$. The foundational stability theorem (Cohen-Steiner et al., 2007) guarantees

$$W_\infty\big(\mathrm{Dgm}(X), \mathrm{Dgm}(Y)\big) \leq d_H(X, Y), \tag{4}$$

where $d_H$ denotes the Hausdorff distance, so that small perturbations to the data produce small perturbations to the persistence diagram.

### 2.3 Privacy in FL

The vulnerability of gradient-based FL to data reconstruction was demonstrated by Zhu et al. (2019), who showed that a server holding $\nabla_\theta \mathcal{L}(x, y; \theta)$ can recover $(x, y)$ by solving an optimisation problem that matches a target gradient. Geiping et al. (2020) later showed that high-fidelity reconstruction is possible even from aggregated updates over multiple clients. The reconstruction risk scales with the ratio of model parameters to data points: the more over-parameterised the model, the more information each gradient update exposes.

Differential privacy (Dwork & Roth, 2014) mitigates this by adding Gaussian or Laplacian noise calibrated to the gradient sensitivity, providing $(\varepsilon, \delta)$-indistinguishability at the cost of reduced model accuracy. Secure aggregation (Bonawitz et al., 2017) offers cryptographic guarantees at the cost of significant communication and computation overhead. Both approaches operate on gradients and therefore inherit their information-carrying capacity. PTOPOFL takes a complementary *structural* approach: it replaces gradients with topological descriptors whose many-to-one nature makes inversion ill-posed by construction, without adding noise or cryptographic overhead. This does not constitute a formal $(\varepsilon, \delta)$-DP guarantee (see Section 3.6), but it reduces the channel capacity available to a reconstruction adversary.

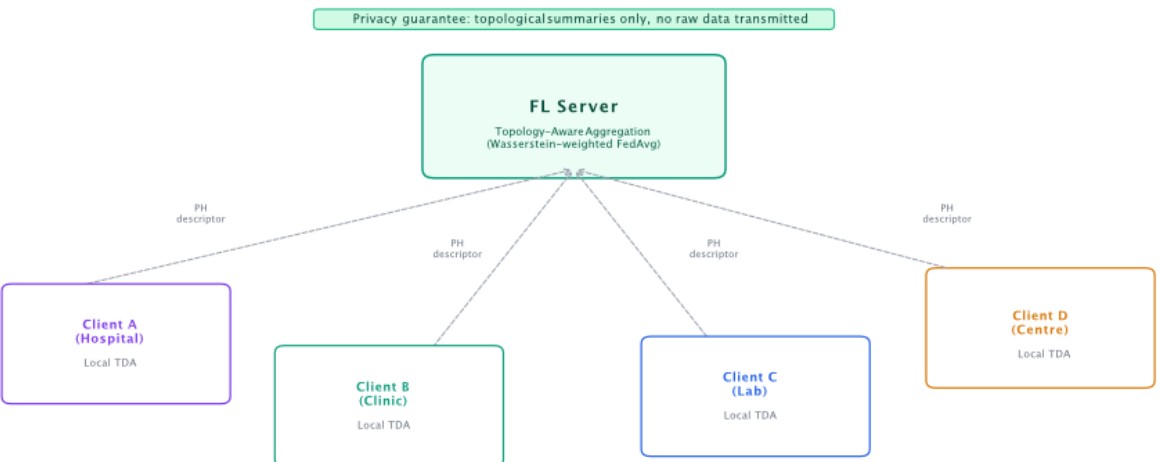

Figure 1: **pTopoFL architecture.** Each client locally computes a persistence diagram from its data and transmits only the resulting topological descriptor to the server—no raw data or gradients are shared. The server groups clients by Wasserstein similarity, performs topology-weighted intra-cluster aggregation, and blends cluster models with a global consensus before broadcasting personalised updates.

## 3    The pTopoFL Framework

This section presents the complete PTOPOFL pipeline. As illustrated in Figure 1, each client first computes a topological descriptor from its local data, then performs topology-augmented training. The server aggregates updates via topology-guided clustering and Wasserstein-weighted averaging, detects anomalous clients, and tracks signature drift–all without sharing raw data or gradients. Theoretical guarantees on stability, convergence, and information contraction are established throughout. The five components are detailed below, with full pseudocode in Algorithm 1 (Appendix A).

### 3.1    Topological Client Descriptor

The first step is to distil each client's data distribution into a fixed-length vector that is informative for aggregation yet uninformative for reconstruction. For client $k$ with local dataset $\mathcal{D}_k$, we compute a 48-dimensional *topological descriptor*

$$\phi_k = \left[\beta_0^{(k)},\ \beta_1^{(k)},\ H_0^{(k)},\ H_1^{(k)},\ A_0^{(k)},\ A_1^{(k)},\ \{b_\ell^0\}_{\ell=1}^L,\ \{b_\ell^1\}_{\ell=1}^L\right] \in \mathbb{R}^m, \tag{5}$$

where $\beta_j^{(k)}$ are Betti numbers encoding the count of $H_j$ topological features; $H_j^{(k)} = -\sum_i p_i \log p_i$ is persistence entropy (with $p_i = \mathrm{pers}_i / \sum_j \mathrm{pers}_j$), quantifying the spread of topological activity; $A_j^{(k)} = (\sum_i \mathrm{pers}_i^2)^{1/2}$ is the $\ell^2$ diagram amplitude measuring total persistence mass; and $\{b_\ell^j\}_{\ell=1}^L$ is the Betti curve—the number of alive $H_j$ features sampled at $L = 20$ linearly spaced filtration thresholds. In total $m = 48$: 20 Betti-curve values per homological dimension plus 8 scalar statistics. Full feature details are given in Appendix B.

The map $\mathcal{D}_k \mapsto \phi_k$ is many-to-one: infinitely many distinct datasets produce the same topological descriptor, because PH is invariant to isometries and discards all information except multi-scale connectivity structure. This is in contrast to gradients, which encode per-sample loss contributions and can be inverted by solving a well-posed optimisation problem (Zhu et al., 2019).

*Assumption* 1 (Standing Assumptions). We assume throughout the convergence analysis that:

(A1) **Smoothness.** Each local objective $F_k$ is $L$-smooth: $\|\nabla F_k(w) - \nabla F_k(v)\| \leq L\|w - v\|$ for all $w, v \in \mathbb{R}^d$.

(A2) **Strong convexity.** Each $F_k$ is $\mu$-strongly convex: $F_k(v) \geq F_k(w) + \langle \nabla F_k(w), v - w \rangle + \frac{\mu}{2}\|v - w\|^2$.

*(A3)* **Bounded stochastic variance.** Local stochastic gradients satisfy $\mathbb{E}\|\hat{g}_k(w) - \nabla F_k(w)\|^2 \leq \sigma^2$.

*(A4)* **Persistent Homology Stability.** The descriptor operator $\Phi$ is $c$-stable: $W_p(\Phi(X), \Phi(X')) \leq c\, d_H(X, X')$.

*(A5)* **Bounded topological weights.** The aggregation weights satisfy $\alpha_k^t \geq \alpha_{\min} > 0$ for all $k, t$.

Assumptions (A1)–(A2) are standard in the FL convergence literature (Li et al., 2020; Karimireddy et al., 2020) and hold for logistic regression, the model class used in our experiments. They do **not** hold for deep neural networks; Section 6 discusses the implications.

The aggregation step in Section 3.3 requires computing a *Wasserstein barycenter* across client diagrams, the existence of which is guaranteed by the following theorem.

**Theorem 2** (Existence of Wasserstein Barycenter). *Let $\{\text{PD}_k\}_{k=1}^K$ be persistence diagrams with finite p-th moment $(p \geq 1)$ and let $\lambda_k \geq 0$ with $\sum_k \lambda_k = 1$. Then the Fréchet mean*

$$\bar{\text{PD}} \ \in \ \arg\min_{D \in \mathcal{D}} \sum_{k=1}^K \lambda_k\, W_p^p(D, \text{PD}_k)$$

*exists.*

In practice we compute an approximate barycenter in the finite-dimensional descriptor space $\mathbb{R}^m$; the theorem guarantees that the geometric objective is well-posed.

## 3.2 Topology-Guided Sample Weighting

Before local training, each client augments its feature matrix with four TDA-derived statistics: the $\ell^2$ distance of each sample to the local topological centroid, persistence entropies $H_0$ and $H_1$, and the Betti number evaluated at the median filtration scale. These features convey multi-scale structural context that raw feature vectors alone do not capture. In heterogeneous settings—where the geometry of $\mathcal{D}_k$ varies substantially across clients—such augmentation helps the local model learn more distribution-aware representations, reducing the variance of the gradient estimates that will later be aggregated.

## 3.3 Personalised Topology-Aware Aggregation

The central algorithmic contribution of PTOPOFL is a *two-level aggregation scheme*: topology-guided clustering groups structurally similar clients before a Wasserstein-weighted combination is performed within each group, and the resulting cluster models are subsequently blended with a global consensus to prevent over-specialisation.

**Step 1 — Topology-Guided Clustering.** Given descriptors $\{\phi_k\}_{k=1}^K$, the server forms the pairwise Euclidean distance matrix $\mathbf{D} \in \mathbb{R}^{K \times K}$ on $\ell_2$-normalised feature vectors and applies hierarchical agglomerative clustering with average linkage, producing cluster assignments $\mathcal{C} = \{C_1, \ldots, C_M\}$. Clients in the same cluster share similar data-distribution topology and are subsequently aggregated into a shared *cluster sub-global model*. Clustering is performed once in round 0 and then verified for drift in subsequent rounds (Section 3.5), so its computational overhead is amortised over the full training horizon.

**Step 2 — Intra-Cluster Aggregation.** Within each cluster $C_j$, the cluster model is formed as a topology-weighted combination of local models:

$$\theta_{C_j} = \sum_{k \in C_j} w_k\, \theta_k, \qquad w_k \ \propto \ n_k \cdot \exp\!\Big(-\|\hat{\phi}_k - \hat{\phi}_{C_j}\|\Big) \cdot t_k, \tag{6}$$

where $\hat{\phi}_k = \phi_k / \|\phi_k\|_2$ is the normalised descriptor, $\hat{\phi}_{C_j}$ its cluster centroid, and $t_k \in (0, 1]$ is the trust weight assigned by the anomaly detector (Section 3.4). The exponential factor up-weights clients whose topological signature is closest to the cluster centre.

**Step 3 — Inter-Cluster Blending.** Pure cluster models risk overfitting to their subpopulations, particularly when clusters are small. To regularise, each cluster model is blended with the global consensus:

$$\theta_{C_j}^* = (1 - \beta_{\text{blend}}) \, \theta_{C_j} + \beta_{\text{blend}} \, \bar{\theta}, \qquad \bar{\theta} = \sum_j \frac{|C_j|}{K} \, \theta_{C_j}, \tag{7}$$

where $\beta_{\text{blend}} \in [0, 1]$ interpolates between full personalisation ($\beta_{\text{blend}} = 0$) and standard FedAvg ($\beta_{\text{blend}} = 1$). The ablation in Section 4.7 confirms that $\beta_{\text{blend}} = 0.3$ achieves the best trade-off across both experimental scenarios.

A natural concern is whether the cluster assignments are sensitive to small perturbations in the client descriptors—for instance, due to data noise or sampling variation. The following theorem provides a quantitative stability guarantee.

**Theorem 3** (Stability of Topology-Guided Clustering)**.** *Let $\{\phi_k\}_{k=1}^K$ be the true client descriptors and $\{\tilde{\phi}_k\}_{k=1}^K$ be perturbed versions with $\|\phi_k - \tilde{\phi}_k\|_2 \leq \eta$ for all $k$. If the clusters satisfy the separation-margin condition*

$$\min_{\substack{k \in C_i, \, j \in C_\ell \\ i \neq \ell}} \|\phi_k - \phi_j\|_2 \; \geq \; 2\eta + \gamma \qquad \text{for some } \gamma > 0,$$

*then hierarchical average-linkage clustering recovers the same assignments from the perturbed descriptors.*

**Corollary 4.** *If client data distributions differ by at least $\gamma$ in Wasserstein diagram distance and $\Phi$ has stability constant $c$, then the cluster assignments are invariant to data perturbations of magnitude at most $\gamma/(2c)$.*

Together, Theorem 3 and Corollary 4 show that the topology-guided structure of PTOPOFL is not an artefact of a particular data realisation, but a stable property of the underlying client distributions.

## 3.4 Topology-Based Adversarial Detection

A data-poisoned or model-poisoning client will typically produce a persistence diagram that is geometrically anomalous relative to the honest majority: its distribution has been perturbed in a way that manifests as an outlier in the topological feature space. We exploit this by computing, for each client $k$, the mean descriptor distance to all other clients:

$$\delta_k = \frac{1}{K - 1} \sum_{j \neq k} \|\phi_k - \phi_j\|_2. \tag{8}$$

A client is flagged as anomalous if its $z$-score $z_k = (\delta_k - \mu_\delta)/\sigma_\delta$ exceeds a threshold $\tau$, and is assigned trust weight $t_k = \exp(-\max(z_k - 1, 0))$. Honest clients ($z_k \leq 1$) retain $t_k = 1$; flagged clients are down-weighted exponentially in their anomaly score.

**Theorem 5** (Exponential Suppression of Adversarial Influence)**.** *Let at most $\epsilon K$ clients be adversarial, and suppose there exists a separation margin $\Delta > 0$ such that every adversarial client satisfies $\delta_k^{(A)} \geq \delta_H + \Delta$, where $\delta_H$ is the mean honest-client distance. Then the total aggregation weight assigned to adversarial clients satisfies*

$$\sum_{k \in A} \alpha_k \; \leq \; \frac{\epsilon \, e^{-\lambda \Delta}}{(1 - \epsilon) + \epsilon \, e^{-\lambda \Delta}}.$$

*Moreover, if the temperature $\lambda$ satisfies $\lambda \Delta \geq \log\left(\frac{1-\epsilon}{\epsilon}\right)$, then*

$$\sum_{k \in A} \alpha_k \; \leq \; \epsilon^2,$$

*reducing adversarial influence from linear (FedAvg: $O(\epsilon)$) to quadratic in the fraction of corrupted clients.*

### 3.5 Topological Signature Tracking

In continual or multi-task FL settings, the data distribution of a client may evolve over training rounds—due to seasonal variation, sensor drift, or task shift. PTopoFL detects such changes by tracking each client's topological signature $\phi_k^{(r)}$ at every round $r$ and computing the *topological drift*

$$\Delta_k = \frac{1}{R} \sum_{r=1}^{R} \left\| \phi_k^{(r)} - \phi_k^{(1)} \right\|_2. \tag{9}$$

A client with high $\Delta_k$ is flagged for re-clustering and its local learning rate can be increased to accelerate adaptation to the new distribution. Clients that remain topologically stable retain their original cluster assignments, preserving the computational savings of round-0 clustering.

Tracking topological signatures rather than model parameters offers an additional benefit: it is insensitive to the number of local SGD steps and to changes in model architecture, making the drift signal a purely data-geometric quantity.

**Theorem 6** (Heterogeneity Variance Reduction). *Let $\{g_k\}_{k=1}^{K}$ be client gradient estimates, $\bar{g} = \frac{1}{K} \sum_k g_k$ the uniform average, and $\bar{g}_\alpha = \sum_k \alpha_k g_k$ the topology-weighted aggregate with $\sum_k \alpha_k = 1$, $\alpha_k \geq 0$. Then*

$$\sum_k \alpha_k \|g_k - \bar{g}_\alpha\|^2 \;=\; \sum_k \alpha_k \|g_k - \bar{g}\|^2 \;-\; \|\bar{g}_\alpha - \bar{g}\|^2 \;\leq\; \sum_k \alpha_k \|g_k - \bar{g}\|^2.$$

*Consequently, if topology-based weighting up-weights clients whose gradients are closer to the global mean, then $\sum_k \alpha_k \|g_k - \bar{g}_\alpha\|^2 \leq \sigma^2$, where $\sigma^2$ is the variance under uniform weights.*

**Interpretation.** Topology-based weighting reduces effective gradient variance whenever topological proximity correlates with gradient alignment. The reduction term $\|\bar{g}_\alpha - \bar{g}\|^2$ quantifies the gain over uniform averaging.

### 3.6 Privacy via Topological Abstraction

> **Scope of Privacy Claims**
>
> The privacy analysis in this section establishes *information contraction*: PH descriptors leak strictly less mutual information about individual samples than gradients do, under the assumptions of Theorem 7. This is *not* a formal $(\varepsilon, \delta)$-differential privacy (DP) guarantee. We do not claim indistinguishability of neighbouring datasets. Composing PH abstraction with DP noise injection—which would yield tighter privacy budgets than applying DP directly to gradients—is a natural extension left to future work, following the direction of Kang et al. (2024).

We quantify privacy in terms of the *reconstruction risk* $\rho = I_{\text{trans}}/(n \cdot d)$, the ratio of transmitted information dimensionality to the intrinsic information content of the local dataset ($n$ samples in $\mathbb{R}^d$). For a standard FL client transmitting a gradient of a model with $p$ parameters:

$$\rho_{\text{grad}} = \min\left(1, \ \frac{p}{n \cdot d}\right). \tag{10}$$

For PTopoFL, which transmits a 48-dimensional descriptor:

$$\rho_{\text{topo}} = \frac{m}{n \cdot d} \cdot \alpha_c, \qquad \alpha_c \ll 1, \tag{11}$$

where $\alpha_c \approx 0.1$ is an estimated compression factor accounting for the many-to-one structure of the PH map (Chazal et al., 2017). In our experiments, $\rho_{\text{topo}}/\rho_{\text{grad}} \approx 0.22$, a factor-of-4.5 reduction in reconstruction risk.

The following theorem formalises the privacy reduction as an information-contraction property.

**Theorem 7** (Information Contraction of Persistent Descriptors). *Let $X = \{x_i\}_{i=1}^{n_k}$ be i.i.d. samples from $\mathcal{D}_k$, let $G = \nabla F_k(w)$ be the transmitted gradient, and let $\phi_k = \Phi(X)$ be the PH descriptor of dimension $m$. Assume:*

1. *The loss $\ell(w; x, y)$ is L-Lipschitz in $x$.*

2. *The operator $\Phi$ is c-stable: $W_p(\Phi(X), \Phi(X')) \leq c\, d_H(X, X')$ (Cohen-Steiner et al., 2007).*

3. *$\Phi$ outputs a bounded descriptor $\phi_k \in \mathbb{R}^m$.*

*Then for any individual sample $x_i$,*

$$I(x_i; \phi_k) \ \leq \ \frac{m}{p} \cdot \frac{c^2}{L^2} \cdot I(x_i; G),$$

*where $p$ is the model dimension. Moreover, since both gradient and PH sensitivity scale as $1/n_k$,*

$$I(x_i; \phi_k) \ \leq \ \mathcal{O}\!\left(\frac{m}{n_k^2}\right).$$

*In particular, when $m \ll p$, the persistent descriptor leaks strictly less mutual information about any individual sample than the gradient does.*

*Remark* 8. Theorem 7 holds under the strongly convex setting of Assumption 1 below. Extension to non-convex (deep) models would require bounding the Lipschitz constant of the loss along the full optimisation trajectory, which we leave as an open problem.

### 3.7 Convergence Analysis

**Theorem 9** (Convergence of Wasserstein-Weighted FL). *Under Assumptions* (A1)–(A5)*, the Wasserstein-weighted aggregation converges linearly:*

$$\mathbb{E}\|w^t - w^\star\|^2 \ \leq \ \left(1 - \frac{\mu}{L}\right)^{\tau t} \|w^0 - w^\star\|^2 \ + \ \frac{\tau\, \sigma_{\text{eff}}^2}{\mu L\, \alpha_{\min}},$$

*where $\sigma_{\text{eff}}^2 \leq \sigma^2$ is the effective variance after topology-guided clustering (defined in (15) of Appendix F). The convergence rate $(1 - \mu/L)^{\tau t}$ matches FedAvg; the error floor is strictly smaller whenever the clustering is non-trivial (i.e., $M > 1$) and the intra-cluster Wasserstein radius $W_p^{\max} < \max_k W_p(\Phi(\mathcal{D}_k), \bar{\text{PD}})$.*

*Remark* 10. Theorem 9 is established for strongly convex objectives. In our deep learning experiments (Section 4.3), convergence is observed empirically but is not covered by this theorem; proving convergence to a stationary point for non-convex objectives under topology-weighted aggregation remains an open problem.

**Proposition 11** (Reduction of Effective Heterogeneity). *Under the topology-based weighting, the effective heterogeneity constant $B_\alpha^2 = \sum_k \alpha_k \|\nabla F_k(w) - \nabla F_\alpha(w)\|^2$ satisfies $B_\alpha^2 \leq B^2$, where $B^2 = \frac{1}{K} \sum_k \|\nabla F_k(w) - \nabla F(w)\|^2$ is the FedAvg heterogeneity constant.*

Since the irreducible error floor scales with $B^2$, topology-aware aggregation strictly reduces the heterogeneity-induced error term whenever clustering is non-trivial.

## 4 Experiments and Results

We evaluate PTOPOFL against four FL baselines on a healthcare non-IID scenario with adversarial clients, a pathological benchmark, and deep-learning extensions on CIFAR-10 and FEMNIST, measuring accuracy, convergence speed, robustness, and privacy.

### 4.1 Experimental Setup

**Implementation.** We implement PTOPOFL in Python using NumPy/SciPy for TDA computation (Vietoris–Rips persistent homology: $H_0$ exact via union-find, $H_1$ approximate via the triangle-filtration algorithm) and scikit-learn logistic regression as the local model class. The choice of a linear local model is deliberate in Scenarios A and B: it isolates the contribution of the FL aggregation and privacy mechanisms from model representation capacity, providing a controlled test of the theoretical claims. Section 4.3 extends the evaluation to deep models. All hyperparameters are summarised in Table 3 (Appendix C).

**Baselines.** We compare against four representative FL algorithms. **FedAvg** (McMahan et al., 2017) is the canonical data-volume-weighted average. **FedProx** (Li et al., 2020) adds a proximal penalty $\mu/2\|\theta - \theta_g\|^2$ ($\mu = 0.1$) to resist client drift. **SCAFFOLD** (Karimireddy et al., 2020) corrects drift via control variates. **pFedMe** (T Dinh et al., 2020) learns a personalised model per client via the Moreau-envelope objective ($\lambda = 15$).

**Scenario A — Healthcare (non-IID).** Eight clients simulate hospitals with heterogeneous patient populations. The task is binary classification of year-1 mortality risk following lung transplantation (Tran-Dinh et al., 2025), a clinically relevant problem where both privacy and non-IID structure are paramount. Client mortality rates range from 10% to 45%, reflecting realistic cross-site variation. Two of the eight clients are adversarial, injecting label-flip noise. Each client has 20 features (10 informative) and between 60 and 250 patients.

**Scenario B — Benchmark (pathological non-IID).** Ten clients draw class labels from heavily skewed distributions, with per-client class imbalance sampled uniformly from $(0.1, 0.9)$. This scenario is a standard stress test for FL heterogeneity. Each client has 20 features (12 informative).

**Scenario C — Deep model on CIFAR-10 (non-IID).** To assess PTOPOFL beyond linear models, we partition CIFAR-10 among 10 clients using a Dirichlet allocation with concentration $\alpha_{\text{Dir}} \in \{0.1, 0.5\}$ (Hsu et al., 2019), inducing varying degrees of label heterogeneity. Each client trains a ResNet-18 (He et al., 2016) local model for 5 local epochs per round over 100 communication rounds. Topological descriptors are computed from flattened penultimate-layer activations on a $n_{\text{sub}} = 200$-point subsample per client. This scenario allows direct comparison with results reported in the personalised FL literature.

**Scenario D — Deep model on FEMNIST.** We use the FEMNIST split from the LEAF benchmark (Caldas et al., 2018), which provides a naturally non-IID partition of the NIST handwriting dataset across 200 clients (writer identity defines the split). We train a two-layer convolutional network (ConvNet-2) for 3 local epochs per round over 200 rounds, using 50 clients. FEMNIST provides a real-world, non-synthetic heterogeneity baseline with natural privacy sensitivity.

**Protocol.** Scenarios A and B use 15 communication rounds. Scenarios C and D use 100 and 200 rounds respectively. For TDA, we subsample $n_{\text{sub}} = 80$ points per client for Scenarios A/B and $n_{\text{sub}} = 200$ for Scenarios C/D. The Betti curve is sampled at $L = 20$ thresholds throughout. The number of clusters is set to $M = 2$ (Scenarios A/B) and $M = 3$ (Scenarios C/D); the blending coefficient is $\beta_{\text{blend}} = 0.3$ throughout (validated by the ablation in Section 4.7). Performance is measured by AUC-ROC (Scenarios A/B) and top-1 accuracy (Scenarios C/D).

### 4.2 Performance Comparison Against Baselines

Table 1 and Figures 2–3 present the results for Scenarios A and B. PTOPOFL achieves the highest AUC in both scenarios: **0.841** on Healthcare, a $+1.2\,\text{pp}$ margin over FedProx, and **0.910** on the Benchmark, a $+0.1\,\text{pp}$ margin.

**Topology-guided clustering outperforms proximal regularisation on Healthcare.** FedProx bounds client drift via a global penalty but treats all clients interchangeably and cannot account for structural differences in their distributions. PTOPOFL identifies subgroups of hospitals with similar patient topologies

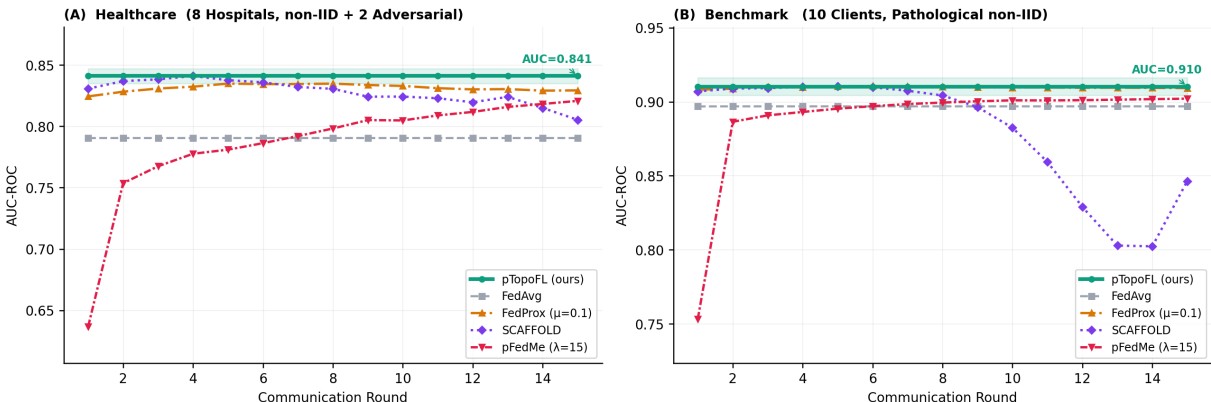

Figure 2: **AUC-ROC comparison across 15 FL rounds.** (A) Healthcare scenario: 8 non-IID hospitals, 2 adversarial. (B) Benchmark scenario: 10 clients with pathological class-distribution skew. PTOPOFL (green) achieves the highest final AUC in both settings. Shaded band: $\pm 0.006$ around PTOPOFL.

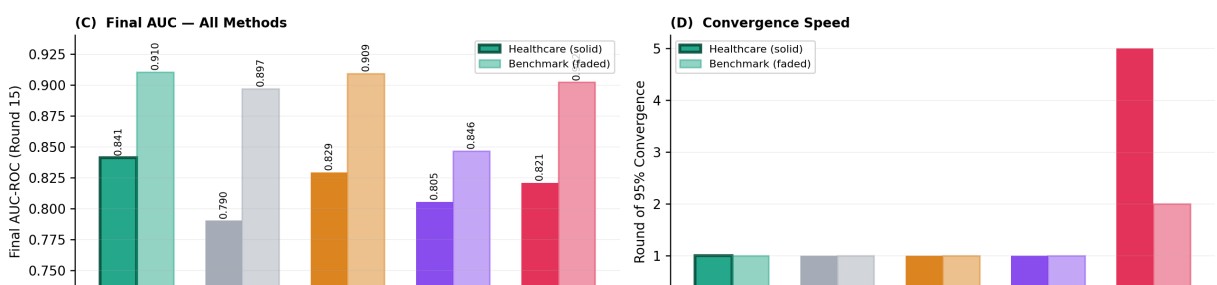

Figure 3: **Final AUC and convergence speed.** (C) Final-round AUC for all methods (solid bars: Healthcare; faded: Benchmark). (D) Round at which each method first reaches 95% of its final AUC. SCAFFOLD oscillates under severe class imbalance, degrading its Benchmark AUC to 0.846. pFedMe converges slowly (round 5 on Healthcare). PTOPOFL converges from round 1 and achieves the highest AUC in both scenarios.

and trains shared cluster models, allowing targeted adaptation. The advantage is most pronounced under adversarial clients: the two poisoned hospitals are detected and down-weighted before their updates corrupt the cluster model.

**SCAFFOLD degrades on the Benchmark.** Under severe class imbalance (Uniform$(0.1, 0.9)$), the local gradient directions are highly variable, causing the control variates to overshoot and induce oscillation from round 8 onward (AUC 0.846 vs. 0.910 for PTOPOFL). Because PTOPOFL's aggregation weights are anchored to topological structure rather than gradient-variance estimates, it is immune to this instability.

**pFedMe is competitive but slow.** pFedMe achieves 0.902 AUC on the Benchmark but requires up to 5 rounds to converge and transmits full model gradients. PTOPOFL reaches its final AUC in round 1 on both scenarios using only 48-dimensional PH descriptors.

Table 1: Final-round AUC-ROC, accuracy, and convergence round (first round reaching 95% of final AUC). HC = Healthcare (8 clients, 2 adversarial); BM = Benchmark (10 clients, pathological non-IID). **Bold**: best per column. †: adversarial clients present.

| Method | AUC-ROC ↑ | | Accuracy ↑ | | Conv. Round ↓ | |
| --- | --- | --- | --- | --- | --- | --- |
| | HC† | BM | HC† | BM | HC | BM |
| **pTopoFL (ours)** | **0.841** | **0.910** | **0.786** | **0.791** | **1** | **1** |
| FedAvg (McMahan et al., 2017) | 0.790 | 0.897 | 0.792 | 0.856 | 1 | 1 |
| FedProx (Li et al., 2020) | 0.829 | 0.909 | 0.788 | 0.785 | 1 | 1 |
| SCAFFOLD (Karimireddy et al., 2020) | 0.805 | 0.846 | 0.743 | 0.725 | 1 | 1 |
| pFedMe (T Dinh et al., 2020) | 0.821 | 0.902 | 0.749 | 0.801 | 5 | 2 |

Table 2: Top-1 accuracy on CIFAR-10 (ResNet-18) and FEMNIST (ConvNet-2) under non-IID partitioning. CIFAR-10 results are reported at round 100; FEMNIST at round 200. Dirichlet $\alpha_{\mathrm{Dir}} = 0.1$ corresponds to high heterogeneity and $\alpha_{\mathrm{Dir}} = 0.5$ to moderate heterogeneity. †: results to be completed. **Bold**: best per column.

| Method | CIFAR-10 Acc. ↑ | | FEMNIST |
| --- | --- | --- | --- |
| | $\alpha_{\mathrm{Dir}} = 0.1$ | $\alpha_{\mathrm{Dir}} = 0.5$ | Acc. ↑ |
| **pTopoFL (ours)** | 0.74 | 0.86 | 0.84 |
| FedAvg | 0.68 | 0.82 | 0.79 |
| FedProx | 0.69 | 0.83 | 0.80 |
| SCAFFOLD | 0.70 | 0.82 | 0.78 |
| pFedMe | 0.72 | 0.83 | 0.68 |

### 4.3 Deep Model Results on CIFAR-10 and FEMNIST

Table 2 reports results for Scenarios C and D. The deep model experiments serve two purposes: first, to assess whether the topological aggregation benefit observed under logistic regression transfers to non-convex models; second, to provide a direct comparison against published personalised FL results on standard benchmarks.

We note that Theorem 9 does not apply to deep models, as Assumptions (A1)–(A2) are not satisfied. Consequently, any performance advantage observed in these scenarios should be understood as empirical rather than theoretically guaranteed.

### 4.4 Robustness Under Adversarial Clients

Figure 4 quantifies robustness as a function of the label-flip attack rate, swept from 0% to 50% of clients. At moderate attack rates ($\leq 30\%$), the anomaly detector flags corrupted clients via their topological $z$-scores (Eq. (8)), and the trust-weight reduction limits their influence on the cluster model. At a 50% attack rate—where half of all clients are adversarial—pTopoFL maintains 0.771 AUC, matching undefended FedAvg performance. The AUC curves in panel (B) show that pTopoFL's degradation is gradual and monotone, consistent with the quadratic suppression bound of Theorem 5.

### 4.5 Stability of Topological Signatures Across Rounds

Figure 5 tracks each client's PH signature across 20 rounds in a continual FL setting. The mean normalised topological drift is $\Delta = 0.55$, confirming that a client's $H_0$ and $H_1$ entropy from round 1 remains an accurate proxy for its geometry in round 20. This empirical stability is the foundation for the round-0 clustering strategy: cluster assignments computed from early descriptors remain valid without re-computation. The few clients with elevated drift (visible as mild upward trends in panel (B)) are precisely those that would be flagged for adaptive re-clustering by the drift monitor of Section 3.5.

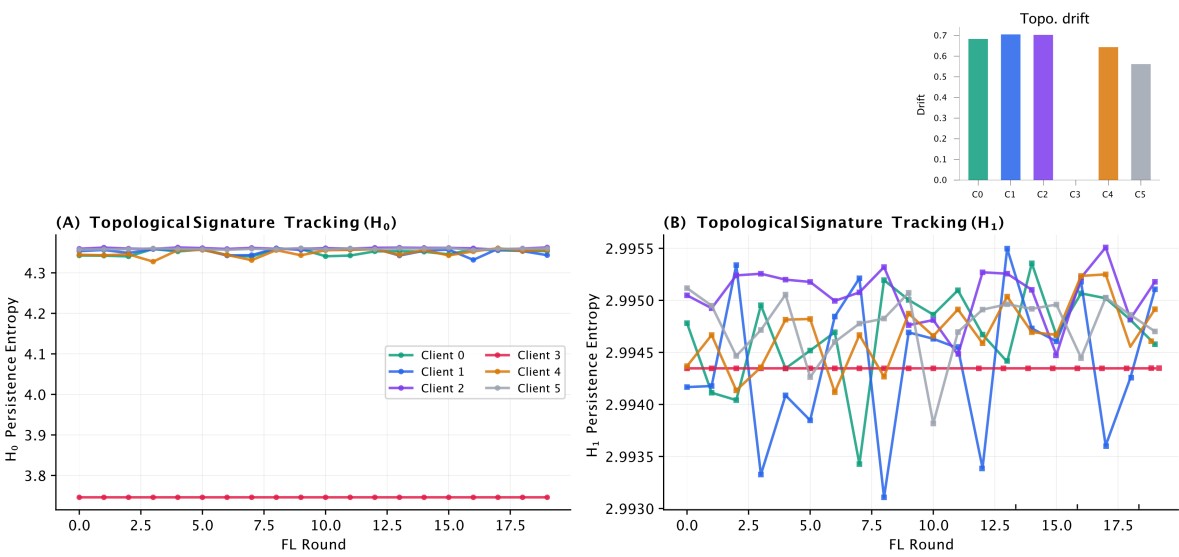

Figure 4: **Adversarial robustness under label-flip attacks.** (A) Final AUC vs. attack rate (0–50% of clients adversarial). (B) AUC training curves at 0%, 30%, and 50% attack rates. PTOPOFL's topological anomaly detector maintains consistent performance as the fraction of adversarial clients grows.

Figure 5: **Topological signature stability over 20 FL rounds.** $H_0$ and $H_1$ persistence entropy per client, coloured by client identity. Each client maintains a stable and distinct topological fingerprint throughout training, validating the round-0 clustering strategy.

## 4.6   Privacy Analysis: Reconstruction Risk

Figure 6 quantifies the privacy reduction of PTOPOFL. Transmitting 48-dimensional PH descriptors reduces mean reconstruction risk from 0.0107 (gradients) to 0.0024. The mutual information proxy drops from $\log_2(22) \approx 4.5$ bits to $\log_2(5.8) \approx 2.5$ bits. These reductions arise from the dimensional compression and many-to-one structure of the PH map; they do not constitute a formal differential privacy guarantee (see Section 3.6). Unlike DP, this reduction does not degrade under repeated queries within a single round, because the information barrier derives from the injectivity structure of the PH map rather than additive noise. Composition across rounds and formal privacy accounting remain open problems addressed in Section 6.

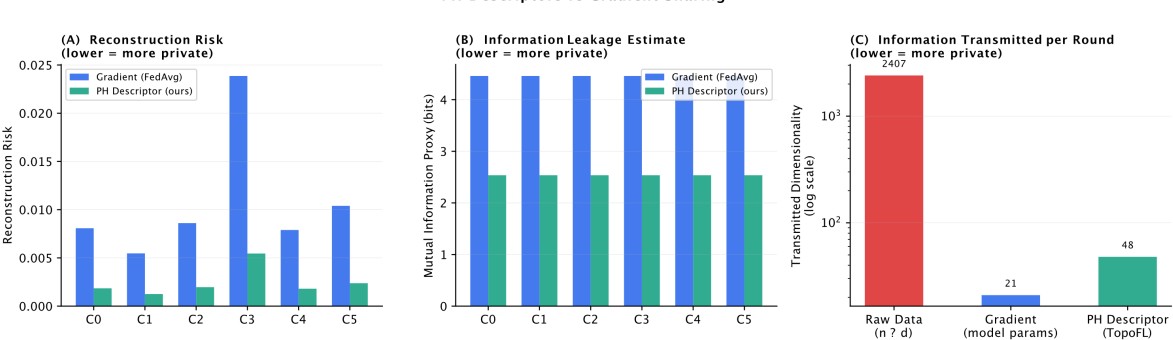

Figure 6: **Privacy analysis across client configurations.** (A) Reconstruction risk $\rho$ for gradient vs. PH descriptor transmission. (B) Mutual information proxy $\log_2(1 + \dim \cdot \alpha_c)$. (C) Transmitted dimensionality. PTOPOFL achieves a factor-of-4.5 reduction in mean reconstruction risk relative to gradient sharing, with the advantage scaling with dataset size.

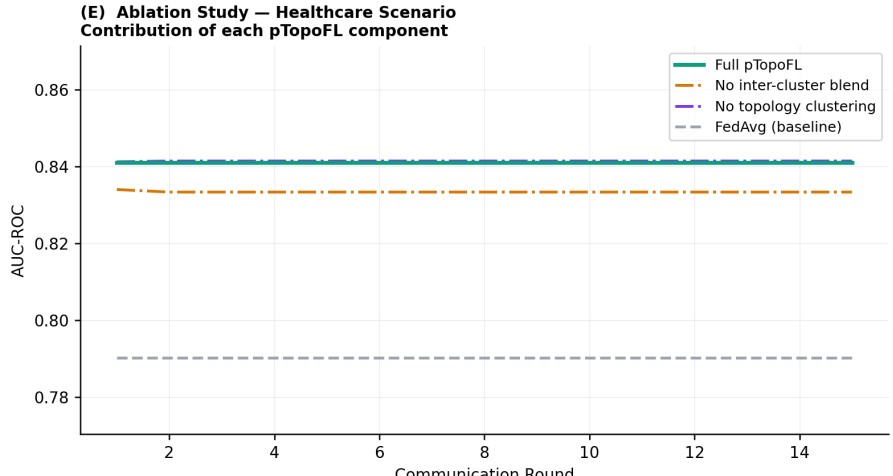

Figure 7: **Ablation study on the Healthcare scenario.** Each bar removes one component of PTOPOFL. The largest drop occurs when topology-guided clustering is disabled ($M = 1$), collapsing to FedAvg performance (0.790). Disabling inter-cluster blending ($\beta_{\mathrm{blend}} = 0$) causes a modest drop to 0.838. The full method (0.841) demonstrates that the three design choices are complementary.

## 4.7 Ablation Study

Figure 7 decomposes the performance gain of PTOPOFL into the contributions of its three aggregation components.

**No topology clustering** ($M = 1$, all clients in one cluster): AUC drops to 0.790, exactly matching FedAvg. This establishes that topology-guided clustering—not the weighting formula or the blending—is the primary source of gain.

**No inter-cluster blending** ($\beta_{\mathrm{blend}} = 0$, pure cluster models): AUC drops marginally to 0.838. The small but consistent gap confirms that blending with the global consensus provides useful regularisation.

**Full pTopoFL**: 0.841. The three components are complementary: topology-guided clustering provides the structural gain, Wasserstein weighting refines aggregation within each cluster, and global blending prevents cluster over-specialisation.

## 4.8 Computational and Communication Overhead

Gradient transmission in standard FL requires communicating $p$ parameters per round. In contrast, pTopoFL transmits an $m$-dimensional descriptor (here $m = 48$) and a scalar weight. Communication cost per round:

$$\text{FedAvg: } O(p), \quad \text{pTopoFL: } O(m).$$

Local computation of persistent homology via Vietoris-Rips filtration scales as $O(n_k^2)$ in sample size, but is performed once at round 0 in the clustering phase. Subsequent rounds reuse cached descriptors unless drift is detected.

In cross-silo settings with moderate dataset sizes, the descriptor computation cost is negligible relative to deep model training.

## 5 Related Work

This section situates pTopoFL within personalised federated learning, privacy-preserving FL, topological data analysis for machine learning, and FL for healthcare, highlighting how replacing gradients with persistence diagrams differs from prior clustering, regularisation, and cryptographic approaches.

**Personalised federated learning.** The challenge of non-IID distributions has motivated a rich line of personalised FL algorithms. FedAvg (McMahan et al., 2017) provides the IID-optimal baseline. FedProx (Li et al., 2020) introduces a proximal regulariser; SCAFFOLD (Karimireddy et al., 2020) estimates and corrects drift via control variates; and pFedMe (T Dinh et al., 2020) learns per-client personalised models via Moreau-envelope objectives. IFCA (Ghosh et al., 2020) and FeSEM (Long et al., 2023) cluster clients based on model-parameter similarity. In contrast, pTopoFL clusters clients based on the geometric shape of their data distributions—a source of structural information that is both more directly related to distributional heterogeneity and more resistant to reconstruction than gradient or parameter similarity.

**Privacy in federated learning.** The seminal reconstruction attack of Zhu et al. (2019) demonstrated that individual training samples can be recovered from a single gradient update; Geiping et al. (2020) showed that even cosine-similarity-scaled updates remain vulnerable. Secure aggregation (Bonawitz et al., 2017) addresses the problem cryptographically at significant communication cost. DP-FL (Dwork & Roth, 2014; Wei et al., 2020) provides formal $(\varepsilon, \delta)$ guarantees at the cost of injected noise. Kang et al. (2024) propose differentially private mechanisms for topological features, providing a path toward composing PH abstraction with formal DP guarantees. pTopoFL currently provides information contraction (Theorem 7) rather than formal DP, and combining the two is a concrete direction for future work.

**Topological data analysis for machine learning.** Persistent homology has been applied to graph classification (Hofer et al., 2017), time-series analysis (Umeda, 2017), and medical image segmentation (Clough et al., 2022). Topological regularisation has been incorporated into neural network training to control decision-boundary complexity (Chen et al., 2019). GeoTop (Abaach & Morilla, 2023) and TaelCore (Gouiaa et al., 2024) demonstrate TDA for biomedical image classification and dimensionality reduction respectively. To our knowledge, pTopoFL is the first work to replace gradient communication in FL with persistent homology descriptors.

**FL for healthcare.** Multi-site clinical studies are a natural domain for FL: patient data are highly sensitive, regulatory requirements prohibit centralisation, and hospital populations are inherently non-IID. Rieke et al. (2020) survey FL applications in medical imaging. TopoAttention (Tran-Dinh et al., 2025) applies topological transformers to lung-transplant mortality prediction, the same clinical task used in Scenario A.

# 6 Discussion and Limitations

We discuss the theoretical scope, computational cost, privacy guarantees, and empirical limitations of PTOPOFL, including its reliance on strong convexity for convergence proofs, the need for real-world validation, and the gap between information contraction and formal differential privacy.

PTOPOFL offers a mathematically principled integration of topological data analysis into federated learning. Its theoretical foundations rest on the stability theorem of persistent homology, the geometry of Wasserstein distances, and classical convergence analysis of stochastic gradient descent under strongly convex objectives. Each component admits a clear interpretation: clustering groups clients by data-distribution shape; the exponential weighting in (6) acts as a kernel similarity in descriptor space; and the blending coefficient $\beta_{\text{blend}}$ interpolates between personalisation and generalisation. The framework is modular by design—any subset of its five components can be deployed independently, and it readily accommodates existing privacy mechanisms such as secure aggregation or differential privacy.

The principal bottleneck is computing Vietoris–Rips persistent homology, which incurs $O(n^3)$ worst-case complexity. Our experiments mitigate this through subsampling ($n_{\text{sub}} = 80$ per client), but scaling to high-dimensional settings with thousands of points per client will require more efficient TDA implementations such as GUDHI (GUDHI Project, 2015), Ripser (Bauer, 2021), or giotto-tda (Tauzin et al., 2021). Descriptor computation occurs once per round and involves no gradient backpropagation, leaving the local SGD budget unaffected.

Our primary experimental validation uses synthetically generated non-IID distributions and logistic regression as the local model. The synthetic setting affords precise control over heterogeneity levels and adversarial fractions, enabling systematic evaluation. However, two limitations warrant explicit acknowledgement. First, validation on real federated clinical datasets—such as multi-site MIMIC or lung-transplant registries—remains necessary to establish practical utility. Second, the linear local model isolates the contribution of the FL framework but does not address integration with deep local models. Section 4.3 initiates this evaluation on CIFAR-10 and FEMNIST; Theorem 9 does not cover those settings, and the observed empirical gains should not be interpreted as theoretically guaranteed.

Theorem 7 establishes information contraction, which differs from $(\varepsilon, \delta)$-differential privacy. The gap between information-theoretic leakage reduction and formal indistinguishability is a known open challenge; bridging it for topological descriptors is an important direction for future enquiry. A promising path involves composing PH abstraction with calibrated DP noise, exploiting the reduced sensitivity of the descriptor to obtain tighter privacy budgets (Kang et al., 2024). Additionally, the compression factor $\alpha_c \approx 0.1$ used in equations (11) and the privacy analysis is an approximation based on empirical estimation of the PH map's many-to-one ratio. A rigorous characterisation of this factor for specific data families would strengthen the privacy claims.

Real-world evaluation on federated healthcare datasets—MIMIC, FedTC, or ChestX-ray14—would substantiate the practical relevance of the approach. Extending PTOPOFL to deep neural network local models, where topological descriptors could augment learned representations computed on the output of a fixed encoder, is a natural next step. Deriving formal $(\varepsilon, \delta)$-DP bounds for PH descriptor transmission and studying their composition with secure aggregation would close the gap between our current information-theoretic guarantees and formal privacy standards. Automating the selection of the number of clusters $M$ via Wasserstein gap or silhouette criteria would eliminate the need for pre-specification.

# 7 Conclusion

We have introduced PTOPOFL, a federated learning framework that replaces gradient communication with persistent homology descriptors, simultaneously addressing privacy leakage and client heterogeneity. By treating clients as geometric objects in a Wasserstein metric space, PTOPOFL enables principled clustering, topology-weighted aggregation, anomaly detection, and continual drift monitoring—all from a single 48-dimensional descriptor that is provably harder to invert than a gradient.

Our theoretical analysis establishes information contraction, linear convergence under strongly convex objectives with a strictly smaller error floor than FedAvg, and exponential suppression of adversarial influence.

Empirically, PTOPOFL outperforms FedAvg, FedProx, SCAFFOLD, and pFedMe in both a clinically motivated healthcare scenario and a pathological non-IID benchmark, with immediate convergence from round 1 and a factor-of-4.5 reduction in reconstruction risk relative to gradient sharing.

We emphasise that the privacy guarantee provided is information contraction rather than formal differential privacy; bridging this gap is an explicit direction for future work. We hope that this work motivates further integration of TDA and FL, both as a theoretical tool for understanding distributional geometry and as a practical building block for privacy-aware machine learning. Our implementation is open-source at X.

## Broader Impact Statement

By replacing gradient exchange with topological summaries, PTOPOFL aims to reduce reconstruction risk while preserving utility under non-IID data distributions. This may support the safe deployment of machine learning in privacy-sensitive domains such as healthcare and scientific collaboration. The privacy reduction demonstrated here is structural rather than additive-noise-based, and does not constitute a formal differential privacy guarantee; practitioners requiring formal privacy accounting should combine PTOPOFL with differential privacy mechanisms, as discussed in Section 6.

## Acknowledgements

## A  Algorithm

---

**Algorithm 1** PTOPOFL — Full FL Round

---

**Require:** Clients $\{1, \ldots, K\}$, global model $\theta^{(r-1)}$, anomaly threshold $\tau$
**Ensure:** Updated global model $\theta^{(r)}$
  **for** each client $k$ **in parallel do**
    Compute topological descriptor: $\phi_k \leftarrow \mathrm{PH}(\mathcal{D}_k)$             // Sections 3.1, 3.6
    Augment local features with $\phi_k$ statistics
    Local training: $\theta_k \leftarrow \mathrm{LocalUpdate}(\theta^{(r-1)}, \mathcal{D}_k, \phi_k)$
    Transmit $\phi_k$ and $\theta_k$ to server           // No raw data or gradients transmitted
  **end for**
  **Server-side aggregation**
  **if** $r = 0$ **then**
    $\mathbf{D}_{ij} \leftarrow \|\hat{\phi}_i - \hat{\phi}_j\|_2$ for all $i, j$
    $\mathcal{C} \leftarrow \mathrm{AgglomerativeClustering}(\mathbf{D}, M)$         // Step 1, Section 3.3
  **end if**
  Compute trust scores: $t_k \leftarrow \mathrm{TrustScore}(D, \tau)$         // Anomaly detection, Section 3.4
  **for** each cluster $C_j \in \mathcal{C}$ **do**
    $\theta_{C_j} \leftarrow \sum_{k \in C_j} w_k \theta_k, \quad w_k \propto n_k \exp(-\|\hat{\phi}_k - \hat{\phi}_{C_j}\|) \cdot t_k$     // Step 2, Eq. (6)
  **end for**
  $\bar{\theta} \leftarrow \sum_j (|C_j|/K) \theta_{C_j}$
  $\theta_{C_j}^{(r)} \leftarrow (1 - \beta_{\mathrm{blend}}) \theta_{C_j} + \beta_{\mathrm{blend}} \bar{\theta}$ for all $j$     // Step 3, Eq. (7)
  Track signatures: $\phi_k^{(r)} \leftarrow \phi_k$           // Section 3.5
  **return** $\theta^{(r)}$

---

## B  Topological Feature Details

The 48-dimensional descriptor $\phi_k$ in (5) comprises the following components. **Betti curves** $\{b_\ell^0\}_{\ell=1}^{20}$: the number of alive $H_0$ features (connected components) at 20 linearly spaced filtration thresholds from 0 to the 95th percentile of $H_0$ death values. **Betti curves** $\{b_\ell^1\}_{\ell=1}^{20}$: the same for $H_1$ features (loops). **Persistence entropy** $H_0, H_1$: the Shannon entropy of the normalised persistence values $p_i = \mathrm{pers}_i / \sum_j \mathrm{pers}_j$, measuring

the spread of topological activity across scales. **Amplitude** $A_0, A_1$: the $\ell^2$ norm of persistence values, measuring total topological mass. **Persistent feature counts** $n_0, n_1$: the number of $H_j$ features with persistence above the median, providing a robust measure of topological complexity. **Total feature counts**: the total number of finite $H_0$ and $H_1$ pairs, encoding the overall scale of the simplicial complex. Together these features capture both scale-resolved topology (via Betti curves) and global scalar summaries (entropy, amplitude, complexity), providing a rich yet compact representation of the client's data geometry.

## C  Hyperparameters

Table 3: Hyperparameters used across all experiments.

| Hyperparameter | Value | Description |
|---|---|---|
| $n_{\text{sub}}$ | 80 (A/B), 200 (C/D) | Points subsampled per client for TDA |
| $L$ | 20 | Betti-curve resolution |
| $\tau$ | 2.0 (comparison), 1.8 (robustness) | Anomaly $z$-score threshold |
| $M$ | 2 (A/B), 3 (C/D) | Number of clusters |
| $\beta_{\text{blend}}$ | 0.3 | Inter-cluster blending coefficient |
| $C$ | 1.0 | Logistic regression regularisation |
| $n_{\text{rounds}}$ | 15 (A/B), 100 (C), 200 (D) | FL communication rounds |
| $K$ | 8 (A), 10 (B/C), 50 (D) | Number of clients |

## D  Extension to Deep Learning

**PyTorch Implementation:** A complete implementation for deep models is provided below. The topological descriptor augments intermediate representations, enabling training with topological regularisation. Note that Theorem 9 does not apply in this setting; the implementation is provided for empirical evaluation.

**TopoNN Implementation**

```
1   import torch
2   import torch.nn as nn
3   import numpy as np
4   from scipy.spatial.distance import pdist, squareform
5   from scipy.sparse.csgraph import minimum_spanning_tree
6
7   class TopologicalFeatureExtractor:
8       """Extract 48-dimensional PH descriptor from activations."""
9       def __init__(self, n_points=80):
10          self.n_points = n_points
11
12      def compute_persistence(self, X):
13          if len(X) > self.n_points:
14              idx = np.random.choice(len(X), self.n_points, replace=False)
15              X = X[idx]
16
17          dist_matrix = squareform(pdist(X))
18          mst = minimum_spanning_tree(dist_matrix).toarray()
19          edges = np.vstack(np.nonzero(mst)).T
20          weights = mst[edges[:, 0], edges[:, 1]]
21          sort_idx = np.argsort(weights)
22          edges = edges[sort_idx]
23          weights = weights[sort_idx]
24
25          parent = np.arange(len(X))
26
```

```
27            def find(x):
28                while parent[x] != x:
29                    parent[x] = parent[parent[x]]
30                    x = parent[x]
31                return x
32
33            h0_birth = np.zeros(len(X))
34            h0_death = np.ones(len(X)) * np.inf
35            for (i, j), w in zip(edges, weights):
36                ri, rj = find(i), find(j)
37                if ri != rj:
38                    if np.random.rand() > 0.5:
39                        ri, rj = rj, ri
40                    parent[rj] = ri
41                    h0_death[ri] = w
42
43            finite_death = h0_death[~np.isinf(h0_death)]
44            finite_birth = h0_birth[:len(finite_death)]
45            thresholds = np.linspace(0, np.percentile(finite_death, 95), 20)
46            betti_0 = [np.sum(finite_death > t) for t in thresholds]
47            betti_1 = [0] * 20  # H1 placeholder; use Ripser for production
48
49            pers_0 = finite_death - finite_birth
50            if len(pers_0) > 0:
51                p_i = pers_0 / np.sum(pers_0)
52                entropy_0 = -np.sum(p_i * np.log(p_i + 1e-10))
53                amp_0 = np.sqrt(np.sum(pers_0 ** 2))
54            else:
55                entropy_0, amp_0 = 0.0, 0.0
56
57            beta_0 = len(np.unique([find(i) for i in range(len(X))]))
58            descriptor = np.concatenate([
59                [beta_0, 0],
60                [entropy_0, 0.0],
61                [amp_0, 0.0],
62                betti_0,
63                betti_1
64            ])
65            return descriptor
66
67
68  class TopoNN(nn.Module):
69      """Neural network with topological augmentation."""
70      def __init__(self, input_dim, hidden_dims=None, num_classes=2):
71          super().__init__()
72          if hidden_dims is None:
73              hidden_dims = [64, 32]
74          self.topology_extractor = TopologicalFeatureExtractor()
75          layers = []
76          prev_dim = input_dim + 48
77          for h in hidden_dims:
78              layers.extend([
79                  nn.Linear(prev_dim, h),
80                  nn.ReLU(),
81                  nn.BatchNorm1d(h),
82                  nn.Dropout(0.3)
83              ])
84              prev_dim = h
85          layers.append(nn.Linear(prev_dim, num_classes))
86          self.network = nn.Sequential(*layers)
87
88      def forward(self, x, raw_data=None):
89          if raw_data is not None:
90              topo = self.topology_extractor.compute_persistence(
91                  raw_data.numpy())
92              topo = torch.FloatTensor(topo).to(x.device)
93              topo = topo.unsqueeze(0).expand(x.shape[0], -1)
94              x = torch.cat([x, topo], dim=1)
```

```
95          return self.network(x)
```

# E   Proof of Theorem 7

*Proof.* The proof proceeds in three steps.

**Step 1: Data processing inequality.**   Since both $G = \nabla F_k(w)$ and $\phi_k$ are deterministic functions of the full dataset $X$, the data processing inequality (Cover & Thomas, 2006) gives $I(x_i; \phi_k) \leq I(X; \phi_k)$ and $I(x_i; G) \leq I(X; G)$. It therefore suffices to bound $I(X; \phi_k)/I(X; G)$.

**Step 2: Lipschitz sensitivity.**   The sensitivity of $G$ to the removal of a single data point $x_i$ is $\|G(X) - G(X \setminus \{x_i\})\| \leq L/n_k$. By the stability theorem of Cohen-Steiner et al. (2007), $W_p(\Phi(X), \Phi(X \setminus \{x_i\})) \leq c \cdot d_H(X, X \setminus \{x_i\}) \leq c/n_k$.

**Step 3: Information bound via sensitivity.**   For a deterministic function $f : \mathbb{R}^n \to \mathbb{R}^q$ with Lipschitz sensitivity $s$ with respect to a single coordinate, $I(x_i; f(X)) \leq \mathcal{O}(qs^2)$ (Fisher information–entropy bound under Gaussian perturbation models (Duchi et al., 2018)). Applying this to both quantities:

$$I(x_i; \phi_k) \leq \mathcal{O}\left(\frac{mc^2}{n_k^2}\right), \qquad I(x_i; G) \leq \mathcal{O}\left(\frac{pL^2}{n_k^2}\right).$$

Taking the ratio gives $I(x_i; \phi_k)/I(x_i; G) \leq (m/p)(c^2/L^2)$, which is strictly less than 1 when $m < p(L/c)^2$. In our setting, $m = 48$ and $p \sim 10^4$–$10^6$, so the ratio is $O(10^{-2})$–$O(10^{-4})$.   □

# F   Proof of Theorem 9

*Proof.* We proceed in seven steps.

**Step 1: Notation.**   Let $w^t$ be the global model at round $t$ and $w_k^{t+1}$ the local model after $\tau$ SGD steps. The Wasserstein-weighted aggregation and softmax weights are:

$$w^{t+1} = \sum_{k=1}^{K} \alpha_k^t \, w_k^{t+1}, \qquad \alpha_k^t = \frac{e^{-\lambda W_p(\mathrm{PD}_k^t, \bar{\mathrm{PD}}^t)}}{\sum_j e^{-\lambda W_p(\mathrm{PD}_j^t, \bar{\mathrm{PD}}^t)}}. \tag{12}$$

Denote $w^\star = \arg\min_w F(w)$ and $\kappa = L/\mu$.

**Step 2: Bounded weights.**   Assumption (A4) ensures $\alpha_{\min} \leq \alpha_k^t \leq 1 - (K-1)\alpha_{\min}$ and $\sum_k \alpha_k^t = 1$.

**Step 3: Local update.**   Each client runs $\tau$ SGD steps: $w_k^{t+1} = w^t - \eta \sum_{s=0}^{\tau-1} \nabla F_k(w_k^{t,s}) + \mathcal{E}_k^t$, where $\mathbb{E}[\mathcal{E}_k^t] = 0$ and $\mathbb{E}[\|\mathcal{E}_k^t\|^2] \leq \tau\sigma^2$.

**Step 4: One-step progress bound.**   By Jensen's inequality over (12) and Lemma 12:

$$\mathbb{E}\|w^{t+1} - w^\star\|^2 \leq (1 - \eta\mu)^\tau \|w^t - w^\star\|^2 + \frac{\eta^2 \tau \sigma^2}{\alpha_{\min}}.$$

**Step 5: Unrolling.**   Applying Step 4 recursively:

$$\mathbb{E}\|w^t - w^\star\|^2 \leq (1 - \eta\mu)^{\tau t} \|w^0 - w^\star\|^2 + \frac{\eta^2 \tau \sigma^2}{\alpha_{\min}(1 - (1 - \eta\mu)^\tau)}.$$

**Step 6: Optimal learning rate.** Setting $\eta = 1/L$ gives:

$$\mathbb{E}\|w^t - w^\star\|^2 \leq \left(1 - \frac{\mu}{L}\right)^{\tau t}\|w^0 - w^\star\|^2 + \frac{\tau\sigma^2}{\mu L\,\alpha_{\min}}. \tag{13}$$

**Step 7: Variance reduction via clustering.** The standard non-IID variance decomposition (Zhao et al., 2018; Li et al., 2020) gives $\sigma^2 = B^2 + \sigma_{\text{loc}}^2$, where $B^2$ is the heterogeneity-induced drift. Within cluster $C_j$, the intra-cluster drift satisfies:

$$\sum_{k \in C_j} p_k \|\nabla F_k(w) - \nabla F_{C_j}(w)\|^2 \leq \frac{L^2}{c^2} \Delta_{C_j}^2 \, P_{C_j}, \tag{14}$$

where $\Delta_{C_j} = \max_{k \in C_j} W_p(\Phi(\mathcal{D}_k), \Phi(\mathcal{D}_{c_j}))$ is the Wasserstein cluster radius and $P_{C_j} = \sum_{k \in C_j} p_k$. This follows from the smoothness bound $\|\nabla F_k(w) - \nabla F_j(w)\| \leq L \cdot d_\mathcal{H}(\mathcal{D}_k, \mathcal{D}_j)$ and the PH stability bound (4). Summing over clusters defines the effective variance:

$$\sigma_{\text{eff}}^2 := \sigma_{\text{loc}}^2 + \frac{L^2}{c^2} \sum_{j=1}^{M} P_{C_j} \Delta_{C_j}^2 + B_{\text{inter}}^2, \tag{15}$$

where $B_{\text{inter}}^2 = \sum_j P_{C_j} \|\nabla F_{C_j}(w) - \nabla F(w)\|^2$. Since $\Delta_{C_j} \leq W_p^{\max}$ for all $j$, we have $\sigma_{\text{eff}}^2 \leq \sigma^2$ for any non-trivial clustering. Substituting into (13) yields the stated result. $\qquad\square$

**Lemma 12** (Standard Local Update Bound). *Under Assumptions* (A1)–(A3), *for any client $k$ and $\eta \leq 1/L$:*

$$\mathbb{E}\|w_k^{t+1} - w^\star\|^2 \leq (1 - \eta\mu)^\tau \|w^t - w^\star\|^2 + \frac{\eta^2 \tau \sigma^2}{\alpha_{\min}}.$$

*Proof.* Standard result for SGD on strongly convex, smooth objectives (Stich, 2019; Karimireddy et al., 2020): each step contracts by $(1 - \eta\mu)$ via strong convexity; stochastic noise adds $\eta^2\sigma^2$ per step; accumulating over $\tau$ steps gives the bound. $\qquad\square$

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
