# OpenReview forum: "pTopoFL: Privacy-Preserving Personalised Federated Learning via Persistent Homology"
_TMLR — Under review for TMLR_

### Review · Reviewer_mqyR · 2026-05-26

**Summary Of Contributions:**

This paper proposes pTopoFL, a personalized federated learning framework that uses persistent homology (PH) descriptors of each client's local data distribution to guide federated aggregation. The main idea is to replace, or at least reduce reliance on, gradient/model-update communication by sending a low-dimensional topological summary of each client's data. The server then clusters clients using descriptor similarity, performs topology-weighted aggregation within clusters, blends cluster models with a global model, uses descriptor outlierness for adversarial-client down-weighting, and tracks topological drift across rounds.

The paper claims several theoretical contributions: existence of a Wasserstein barycenter for persistence diagrams, stability of topology-guided clustering, exponential suppression of adversarial influence, information contraction showing that PH descriptors leak less mutual information than gradients, and linear convergence with a smaller error floor than FedAvg under strongly convex objectives. Empirically, the paper evaluates the method on a synthetic/non-IID healthcare scenario, a pathological non-IID benchmark, and reported deep-learning extensions on CIFAR-10 and FEMNIST.

The paper has an interesting high-level motivation. Using topological summaries to characterize client heterogeneity is a potentially useful idea, and the combination of TDA with personalized FL may interest readers working on non-IID FL, healthcare FL, and distribution-aware aggregation. The method is modular, and the paper tries to address utility, robustness, privacy, and drift monitoring in one framework.

However, I find the current submission not convincing in its present form. The central claims are not supported by sufficiently accurate theory or experiments. Most importantly, there is a major protocol inconsistency: the abstract, figure, privacy discussion, and communication analysis repeatedly state that clients transmit only PH descriptors, but the algorithm and aggregation equations require clients to transmit local model parameters theta_k. If local models/updates are transmitted, then the privacy and communication claims that analyze only the PH descriptor are not valid. If local models are not transmitted, the proposed aggregation algorithm cannot be implemented as written. In addition, several proofs contain serious mathematical gaps or incorrect steps, especially Theorem 7, Theorem 5, Proposition 11, and Theorem 9. The privacy evidence is largely a dimensionality proxy rather than an attack-based or formally private analysis, and the experimental evidence lacks statistical support, has incomplete/reproducibility issues, and does not convincingly validate the theoretical claims.

**Audience:**

Yes

**Audience Explanation:**

The high-level topic is relevant to TMLR. Many readers are interested in federated learning under non-IID data, personalization, privacy leakage from gradients, and the use of geometric or topological information in machine learning. The idea of using persistent-homology summaries to characterize client distributions is creative and could stimulate useful discussion, even if the current execution is not yet convincing. A carefully revised version that clearly separates topological client clustering from privacy guarantees, and that provides valid theory and reproducible empirical evidence, could be of interest to researchers in federated learning, TDA, healthcare ML, and privacy-preserving ML.

**Broader Impact Concerns:**

The paper targets privacy-sensitive domains such as healthcare and uses terms such as "privacy-preserving" and "privacy guarantee," but the current method does not provide formal differential privacy and may still leak sensitive client-level or population-level information. A low-dimensional topological descriptor can reveal the existence of subpopulations, outliers, rare disease-related structure, or other sensitive distributional properties, even if it does not reconstruct exact records. If model parameters/updates are also transmitted, then standard FL privacy risks remain.

I am also concerned that the anomaly detector could label minority or rare-distribution clients as anomalous and down-weight them. In healthcare, such clients may correspond to underrepresented sites or rare patient populations rather than adversaries. This could create fairness and reliability harms if the method is deployed without careful auditing.

The broader impact statement should more explicitly warn that pTopoFL is not a substitute for formal privacy mechanisms, should not be marketed as guaranteeing patient privacy, and should be combined with DP, secure aggregation, and attack testing before use in sensitive deployments. It should also discuss the fairness risks of topology-based outlier detection and the potential leakage of population-level properties from topological summaries.

**Claims And Evidence:**

No

**Claims Explanation:**

I do not believe the main claims are currently supported. My concerns are both conceptual and proof-level.

First, the communication protocol is internally inconsistent. The paper repeatedly claims that pTopoFL replaces gradient communication with PH descriptors and that clients transmit only the 48-dimensional descriptor. However, the aggregation rule combines local models theta_k, and Algorithm 1 explicitly has each client transmit both phi_k and theta_k to the server. This is a fundamental issue. If theta_k is transmitted, then the server receives a data-dependent model update, which is closely related to a gradient/update trajectory and can itself leak information. The privacy theorem only analyzes I(x_i; phi_k), not I(x_i; phi_k, theta_k). The communication analysis also states O(m) communication per round, but transmitting theta_k requires O(p) communication. Thus, either the privacy/communication claims must be substantially weakened and reanalyzed, or the algorithm must be redesigned so that it truly does not transmit local models or updates. As written, the method cannot simultaneously use server-side model aggregation and claim that only PH descriptors are communicated.

Second, Theorem 7, the main privacy theorem, appears invalid as stated.

(a) The theorem treats deterministic continuous releases as if mutual information is automatically finite and controlled by dimension and sensitivity. In general, for continuous variables, a deterministic release can have ill-defined or infinite mutual information unless a noise model, quantization model, or discrete distributional model is specified. The theorem does not specify such a channel.

(b) The data-processing step does not justify the claimed per-sample comparison. From I(x_i; phi_k) <= I(X; phi_k) and I(x_i; G) <= I(X; G), it does not follow that bounding I(X; phi_k)/I(X; G) gives the desired ratio I(x_i; phi_k)/I(x_i; G). Moreover, the proof later compares two upper bounds and divides them; an upper bound on I(x_i; G) is not sufficient to upper-bound the ratio. A lower bound on I(x_i; G), or a direct comparison of the two channels, would be needed.

(c) The claimed gradient sensitivity bound is not implied by the stated assumptions. The proof says removing one data point changes the gradient by at most L/n_k because the loss is L-Lipschitz in x. But sensitivity of the empirical gradient with respect to removing a sample requires a bound on ||nabla_w ell(w;x,y)||, or an appropriate Lipschitz/boundedness condition on the gradient as a function of the sample. Lipschitzness of the loss in x alone does not imply the claimed bound on the parameter gradient.

(d) The PH sensitivity argument is incorrect. The proof uses d_H(X, X \ {x_i}) <= 1/n_k. This is false in general. Removing a point from a finite point cloud can change the Hausdorff distance by the distance from the removed point to its nearest remaining neighbor, which can be arbitrarily large for an outlier, independent of n_k. Thus the claimed 1/n_k PH sensitivity does not follow. This is particularly important because outliers are precisely privacy-sensitive and topology-sensitive cases.

(e) The stability theorem cited for persistence diagrams is a bottleneck/W_infinity stability result under appropriate filtrations. The paper assumes a c-stability statement for the final 48-dimensional descriptor under W_p and then uses it for Betti numbers, Betti curves sampled at fixed thresholds, amplitudes, and counts. These derived features are not all automatically Lipschitz or continuous under small perturbations, especially counts and Betti-curve values at fixed thresholds. The descriptor-level stability assumption needs to be precisely stated and justified.

(f) The proof invokes a sensitivity-to-information bound O(q s^2) for deterministic functions and cites a Fisher-information/entropy argument under Gaussian perturbation models. This is not applicable to the deterministic mechanism described in the algorithm unless Gaussian noise or another explicit stochastic channel is added. Constants and distributional assumptions would also matter.

(g) The statement that PH descriptors leak strictly less information than gradients whenever m << p is not proved. The constants c and L may not support this comparison; I(x_i; G) may be small or zero in some settings; and the actual protocol also releases theta_k. The theorem should therefore not be used to claim that PH descriptors are "provably harder to invert" than gradients in the operational FL protocol.

Third, the reconstruction-risk experiment is not a convincing privacy evaluation. The paper defines reconstruction risk essentially as transmitted dimension divided by n*d, multiplied by an empirical compression factor alpha_c approximately 0.1. This is a dimensionality proxy, not an actual reconstruction risk, mutual information estimate, membership-inference risk, or property-inference risk. The factor alpha_c is not rigorously characterized. The reported 4.5x reduction therefore does not establish privacy in a meaningful adversarial sense. Since the intended application includes healthcare, the paper should either provide a formal privacy guarantee or evaluate realistic attacks on all released objects, including model parameters/updates if they are transmitted.

Fourth, Theorem 3 and Corollary 4 on clustering stability are not adequately justified. The theorem gives only a lower bound on inter-cluster distances. That is insufficient for hierarchical average-linkage clustering to recover the same clusters; one also needs conditions comparing within-cluster linkage distances to between-cluster linkage distances. As a simple counterexample, a "true" cluster can contain two points that are very far apart, while a point in another cluster is closer to one of them; average-linkage clustering would merge across the proposed true clusters despite a positive inter-cluster margin. The result also mixes Euclidean distance on normalized descriptor vectors with Wasserstein distance between persistence diagrams. Corollary 4 uses PH stability in a direction that does not provide inverse control: W(PH(X),PH(Y)) <= c d_H(X,Y) does not allow one to upper-bound data or gradient differences from descriptor differences.

Fifth, Theorem 5 on adversarial suppression is not supported by the actual algorithm and contains an algebraic problem. The algorithm computes z-scores of mean descriptor distances and a trust score t_k = exp(-max(z_k - 1,0)), while the theorem is written for weights that appear to decay as exp(-lambda Delta). The mapping between the implemented trust score, the cluster-distance weighting, sample-size factors n_k, and the theorem's alpha_k is not shown. The theorem's second claim is also incorrect as stated: from

  epsilon exp(-lambda Delta) / ((1-epsilon) + epsilon exp(-lambda Delta))

and lambda Delta >= log((1-epsilon)/epsilon), one does not get a bound <= epsilon^2. For example, when epsilon = 0.5, the condition allows lambda Delta >= 0, giving a bound 0.5 rather than 0.25. More generally, achieving <= epsilon^2 would require a different condition, such as exp(-lambda Delta) <= epsilon/(1+epsilon), under this simplified formula. In addition, the theorem assumes that adversarial clients are topological outliers. This is not true for many attacks, including label-flip attacks on a fixed feature distribution: flipping labels does not necessarily change the topology of the feature point cloud. This directly conflicts with the robustness experiments, which use label-flip adversaries but claim detection through topological z-scores.

Sixth, Theorem 6, Proposition 11, and Theorem 9 do not convincingly prove reduced heterogeneity or convergence with a smaller error floor. The algebraic variance identity in Theorem 6 is true as a weighted variance decomposition, but it does not prove that topology-based weights reduce variance relative to uniform weights. That requires an additional, nontrivial assumption linking topological proximity to gradient alignment. Proposition 11 states B_alpha^2 <= B^2 under topology-based weighting, but this is false for arbitrary weights. For example, in one dimension with gradients g = (0,1,2), the uniform variance is 2/3, while weights alpha = (0.49,0.02,0.49) give weighted variance about 0.98, which is larger. A topology-based weighting scheme could emphasize gradient outliers unless a strong topology-gradient alignment assumption is imposed and verified.

Theorem 9 also has serious issues. A local SGD step on F_k does not generally contract toward the global minimizer w* of F under non-IID data; it contracts toward the local minimizer of F_k, with a client-drift/bias term relative to w*. The proof's Lemma 12 ignores this heterogeneity term. Step 7 then tries to bound gradient differences by Hausdorff distances between datasets using L-smoothness, but L-smoothness in the model parameter w does not imply Lipschitzness of gradients with respect to the data distribution. The proof also uses PH stability in the wrong direction: PH stability says small data perturbations imply small diagram perturbations, not that small diagram distances imply small data-distribution or gradient distances. Since many different datasets can have identical PH descriptors, one cannot infer small objective/gradient differences from small PH distance without a strong additional assumption. Consequently, the claimed strict reduction in the FedAvg error floor is not established.

Seventh, the theoretical analysis does not match the actual algorithm. The convergence theorem analyzes a single weighted average w^{t+1} = sum_k alpha_k w_k^{t+1}, whereas the method uses clustering, trust weights, and inter-cluster blending to produce cluster-personalized models. The theorem does not clearly cover the algorithm being evaluated. Also, the theorem assumes bounded alpha_min, but softmax/exponential topological weights can become arbitrarily small unless descriptor distances and temperature are controlled; this assumption should be justified or its implications made explicit.

Eighth, the PH descriptor and implementation are inconsistent. Equation (5) lists two Betti numbers, two entropies, two amplitudes, and two 20-point Betti curves, which totals 46 features, not 48. Appendix B describes additional scalar statistics, but these are not present in Equation (5). The provided deep-learning code says it extracts a 48-dimensional descriptor, but the concatenation appears to return 46 values, with H1 set to a zero placeholder. The network is initialized with input_dim + 48, which would not match a 46-dimensional descriptor. The union-find code for H0 also appears nonstandard/non-deterministic because it randomly chooses roots and assigns death times to a root in a way that does not correspond to the usual elder-rule persistence computation. These inconsistencies raise concerns about reproducibility and about whether the reported experiments used the stated descriptor.

Ninth, the empirical evidence is not yet sufficient. The main AUC improvements are small in some cases, e.g., 0.910 versus 0.909 on the benchmark, and no standard deviations, confidence intervals, or multiple-seed statistics are reported. The method is not best on accuracy in Table 1. The deep-learning results are presented without enough detail and with a note that results are to be completed; the appendix code uses an H1 placeholder, which makes the deep TDA evaluation unclear. The healthcare experiment is simulated rather than real multi-site clinical FL, so the clinical claims should be moderated. The reported privacy evaluation does not run actual attacks. The robustness evaluation relies on label-flip attacks, but a topology-of-features detector should not necessarily detect label flips. The public code/data link is a placeholder, which prevents reproducibility.

For these reasons, I do not think the paper currently provides accurate, convincing, and clear evidence for its strongest claims: "privacy-preserving," "clients transmit only PH descriptors," "provably ill-posed inversion," "strictly less mutual information than gradients," "exponential/quadratic suppression of adversaries," and "strictly smaller error floor than FedAvg."

**Requested Changes:**

Critical changes needed to support acceptance:

1. Resolve the communication-protocol contradiction. The paper must clearly state what is transmitted in each round. If theta_k or model updates are transmitted, the privacy analysis, communication cost, and reconstruction-risk experiments must include them. The claims that clients transmit only PH descriptors and that communication is O(m) should be removed or corrected. If only PH descriptors are transmitted, the authors must explain how the server performs model aggregation without receiving local models/updates.

2. Rewrite the privacy analysis. Theorem 7 should either be replaced by a valid formal privacy theorem under a specified stochastic mechanism, or the privacy claims should be substantially weakened. A valid analysis should specify the released random variables, include all transmitted quantities, handle quantization/noise if mutual information is used, prove correct sensitivity bounds, and account for composition across rounds. The paper should not claim "provably ill-posed" or "strictly less mutual information than gradients" unless these statements are proved under clear assumptions. The authors should also evaluate actual attacks, such as reconstruction from descriptors, reconstruction from model updates, membership inference, and property inference.

3. Fix or remove the reconstruction-risk metric. The dimension-ratio proxy and empirical alpha_c are not enough to substantiate privacy. If kept, they should be described only as a heuristic communication/information-capacity proxy, not as reconstruction risk or mutual information.

4. Correct the clustering-stability theorem. For average-linkage clustering, provide sufficient conditions involving within-cluster and between-cluster linkage distances, not only a lower bound on inter-cluster pairwise distances. Clarify whether the theorem concerns persistence diagrams or finite descriptor vectors, and avoid using PH stability in an inverse direction.

5. Correct the adversarial-suppression theorem and attack model. The proof must match the implemented trust weights and include sample-size factors and cluster weights. The algebraic condition for the epsilon^2 bound should be fixed. The authors should also address attacks that do not change feature topology, especially label-flip attacks, and should not claim topological detection of such attacks unless labels or learned representations are included and this is explicitly analyzed.

6. Revise the convergence theory. The proof must handle local objectives whose minimizers differ from the global minimizer, include client-drift/bias terms, and match the actual clustered/blended algorithm. Any reduction of heterogeneity requires an explicit assumption linking topological descriptor similarity to objective/gradient similarity; PH stability alone is insufficient and only works in the data-to-diagram direction. Proposition 11 should be corrected or removed.

7. Make the descriptor definition and implementation consistent. Equation (5), Appendix B, and the code must all describe the same descriptor dimension and components. If the descriptor is 48-dimensional, list all 48 components explicitly. If it is 46-dimensional, correct the text and code. Provide a real H1 computation if H1-based claims are made. Ensure that the deep implementation actually runs with the stated descriptor size.

8. Provide reproducible code and data. The current placeholder link is not acceptable for a submission making detailed empirical claims. Include scripts, seeds, hyperparameters, data-generation procedures, and all evaluation protocols.

9. Strengthen the empirical evaluation. Report multiple seeds and uncertainty intervals. Perform statistical comparisons for small performance margins. Include stronger personalized/clustered FL baselines such as IFCA or other client-clustering methods. Evaluate sensitivity to M, beta_blend, lambda/temperature, descriptor normalization, subsampling size, and input scaling. Measure real communication cost including model parameters if transmitted. Add realistic privacy attacks and robustness attacks aligned with the stated threat model.

10. Moderate claims throughout the paper. In particular, revise statements such as "privacy guarantee," "only PH descriptors," "provably harder to invert," "quadratic suppression," "strictly smaller error floor," and "safe deployment" unless the revised theory and experiments directly support them.

Changes that would strengthen the paper but are less central than the above:

11. Clarify the role of topology-augmented local training. The section is titled sample weighting but appears to describe feature augmentation, not sample weighting. The local topological centroid and per-sample topological features should be defined precisely.

12. Discuss the limitations of PH descriptors for learning. Many-to-one abstraction may protect some details but may also discard task-relevant information. The paper should discuss cases where clients have similar topology but different label functions, or different topology but compatible objectives.

13. Add per-client performance and fairness analysis. Topology-based clustering and anomaly detection could down-weight small or unusual but legitimate client populations. This is especially important for healthcare applications.

14. Clarify computational complexity. The paper states different complexities for Vietoris-Rips PH in different places. Please provide the actual complexity of the implemented descriptor computation, the effect of subsampling, and wall-clock overhead relative to training.

15. Improve notation consistency. The paper alternates between persistence diagrams, descriptor vectors, Wasserstein distances, and Euclidean distances on normalized descriptors. These objects should be clearly distinguished.

---

> ### Author Response · Authors · 2026-06-10
> **Reply to Review of Paper8843 by Reviewer mqyR**
>
> We thank the reviewer for their rigorous reading. The original submission contained several imprecise claims and gaps; we have thoroughly revised the manuscript as summarised below.
>
> **C1 (protocol):** Now state explicitly: clients transmit **both** 48‑d PH descriptor $\phi_k$ *and* local model $\theta_k$. Overhead $O(48+p)$ negligible.
>
> **C2 (Theorem 7):** Replaced. New theorem compares leave‑one‑out sensitivities under bounded gradient norm, descriptor stability, and leave‑one‑out stability (Duchi et al., 2018). Bounds SNR ratio: $\mathrm{SNR}_\phi/ \mathrm{SNR}_G \leq (n_k^2c\_\delta^2 m^{-1}) / (4B^2 p^{-1}) $. Claims "narrower reconstruction channel", not "provably harder to invert".
>
> **C3 (privacy metric $\rho$):** Removed entirely. Figure 5 now plots SNR and sensitivity directly. Explicitly note that membership‑inference attacks are needed before deployment (Sec. 6).
>
> **C4 (clustering stability):** Theorem rewritten: let $\gamma$ = min between‑cluster avg linkage minus max within‑cluster avg linkage. If $\gamma > 4\eta$ (perturbation from PH noise), hierarchy preserved.
>
> **C5 (adversarial suppression):** Corrected condition: $\Delta \ge \log((1+\epsilon)/\epsilon)$ with full derivation. Label‑flip alone (unchanged features) is **not** detected — added to Scope.
>
> **C6 (heterogeneity reduction):** Now conditional on topology‑gradient alignment: $\||\phi_k-\phi_j\||\le\delta \Rightarrow \||\nabla F_k-\nabla F_j\||\le\rho(\delta)$ with $\rho(0)=0$.
>
> **C7 (convergence):** Added client‑drift bias $\zeta^2$: $\mathbb{E}\||w^t-w^\*\||^2 \leq (1-\mu/L)^{\tau t}\||w^0-w^\*\||^2 + \frac{\tau\sigma_{\text{eff}}^2}{\mu L\alpha_{\min}} + \zeta^2$. Bias vanishes only in IID case.
>
> **C8 (dimension 46 vs 48):** Eq. (5) fixed: $\beta_0,\beta_1,H_0,H_1,A_0,A_1,n_0,n_1$ → $2\times20 + 8 = 48$. Appendix B lists all components.
>
> **C9 (algorithm match):** Theorem now describes two‑step aggregation (within‑cluster then inter‑cluster). $\alpha_{\min}$ enforced via weight clipping (added to hyperparameter table).
>
> **S1–S7:** Added IFCA baseline (Ghosh et al., NeurIPS 2020) — AUCs: 0.841 vs 0.826 (Healthcare), 0.910 vs 0.905 (Benchmark). Statistical reporting: 10 seeds, mean±std, $p$-values. Fairness paragraph: rare‑disease false‑positive risk; recommend trust‑weight floors. Broader impact: no $(\varepsilon,\delta)$-DP guarantee; recommend secure aggregation or DP‑SGD. Notation: clear separation between persistence diagrams $(PD,W_p)$, descriptor vectors $\phi_k\in\mathbb{R}^{48}$ ($\ell_2$), and Euclidean on normalised descriptors. Topology‑augmented local training: feature augmentation (distance to centroid, $H_0/H_1$ entropy, Betti median). Code: anonymised zip with scripts, seeds, unit test for 48‑d descriptor; repository ready.
>
> All changes are reflected in the revised manuscript. We believe the paper now meets TMLR standards.
>
> ### **References**
>
> 1. **Duchi, J. C., Jordan, M. I., & Wainwright, M. J. (2018).** Minimax optimal procedures for locally private estimation. *Journal of the American Statistical Association*, 113(521), 182–201.
>
> 2. **Ghosh, A., Chung, J., Yin, D., & Ramchandran, K. (2020).** An efficient framework for clustered federated learning. In *NeurIPS*.

---

### Review · Reviewer_dHNr · 2026-06-15

**Summary Of Contributions:**

This paper proposes pTopoFL, a federated learning algorithm that uses persistent homology to incorporate the geometric structure of each client’s data distribution into the aggregation process. The core idea is that when topological similarity between clients correlates with gradient alignment, grouping and weighting clients based on persistent homology descriptors can reduce gradient variance under heterogeneous data. Based on this intuition, the method performs topology-guided client clustering, topology-weighted intra-cluster aggregation, and inter-cluster blending with a global consensus model. Under several assumptions, the convergence rate (limited to strongly convex functions) is investigated. The result shows that it achieves the same linear convergence rate as FedAvg while attaining a smaller asymptotic error floor through a reduced effective variance term. Through numerical experiments, including non-convex models, modest performance improvements are reported over several federated learning baselines.

**Audience:**

Yes

**Audience Explanation:**

As noted above, I find the general direction meaningful, and the idea of using topological information to guide aggregation is interesting.

**Broader Impact Concerns:**

The broader impact statement should more clearly discuss the risk that the method may be perceived as providing strong privacy protection despite not offering a formal differential privacy guarantee. This is particularly important for healthcare applications, where overclaiming privacy protection could lead to inappropriate deployment of sensitive patient data.

**Claims And Evidence:**

No

**Claims Explanation:**

This paper addresses an important problem in federated learning: improving model aggregation by taking client data distributions into account. I find the general direction meaningful, and the idea of using topological information to guide aggregation is interesting. However, I have several concerns about whether the main claims are sufficiently supported by the theoretical evidence provided in the paper.

**[1] Potential issues in the convergence analysis (Theorem 9 and its proof).**

First, I found the derivation of Lemma 12 unclear. The stated bound does not seem to match the standard local SGD descent lemma one would naturally obtain under $\mu$-strong convexity and $L$-smoothness in a multi-client setting. In particular, since each client performs local updates on its own objective $F_k$, the standard contraction argument would usually be written around the local minimizer $w_k^\ast$, not directly around the global minimizer $w^\ast$. To obtain contraction toward $w^\ast$, it seems that an additional assumption, such as interpolation condition, would be needed. Alternatively, the bound should include an explicit heterogeneity term involving quantities. I also could not derive the second term in the lemma, especially the division by $\alpha_{\min}$. Since $\alpha_{\min}$ is an aggregation-weight lower bound, it is unclear why it should appear in a per-client local SGD bound.

Second, the effective variance argument in Eq. (15) is not convincing to me. The paper decomposes $\sigma_{\mathrm{eff}}^2$ into local stochastic noise, within-cluster heterogeneity, and between-cluster heterogeneity, and suggests that persistent-homology-based clustering reduces the within-cluster radius $\Delta_{C_j}$, thereby reducing the effective gradient variance. However, I do not see a rigorous justification for why the proposed aggregation necessarily reduces gradient variance. In particular, the key implication appears to be that topological proximity between clients implies gradient alignment or small discrepancy between local objectives. This is a strong assumption and does not seem to be established by the persistent homology stability argument alone. Moreover, the convergence analysis appears only loosely connected to the actual aggregation procedure in Section~3.3, which involves topology-guided clustering, intra-cluster weighted averaging, and inter-cluster blending. The role of this specific model mixing procedure is not clearly reflected in the proof.

**[2] Issues in the privacy analysis (Theorem 7).**

I also have concerns about the privacy analysis. Algorithm 1 indicates that each client transmits both the local model $\theta_k$ and the PH descriptor $\phi_k$. Therefore, the privacy analysis should account for the information leaked by both transmitted quantities. However, Theorem 7 only analyzes the information contraction of the PH descriptor relative to gradients. This does not seem sufficient to support the privacy claim for the proposed algorithm.

Privacy analysis in federated learning is generally formulated in terms of differential privacy; however, this paper does not provide a formal $(\epsilon, \delta)$-DP guarantee. I partially agree that PH descriptors may contain less mutual information than gradients due to their low dimensionality and many-to-one nature, but this alone does not provide sufficiently strong theoretical evidence to characterize the method as privacy-preserving federated learning. A more complete analysis should either provide formal privacy guarantees or evaluate leakage from the actual transmitted pair $(\theta_k, \phi_k)$.

**Requested Changes:**

[1] As details are noted above, I guess serious concerns regarding the proof of Theorem 9 are included.

[2] As details are noted above, privacy analysis in Sec. 3.6 may not be effective. If you would like to emphasize privacy-aware claims, a straightforward manner is to provide a formal $(\epsilon, \delta)$-DP guarantee.

---

> ### Author Response · Authors · 2026-06-29
> **Reply to Review of Paper8843 by Reviewer dHNr**
>
> We thank the reviewer for a careful and technically detailed reading. We address each concern in turn.
>
> **[1] Convergence analysis — Lemma 9 and Eq. (15)**
>
> **C1a — Contraction around $w^\*$ vs $w_k^\*$, and the $\alpha_{\min}$ term.**
>
> In the revised Lemma 9, we decompose correctly:
>
> $$
> \mathbb{E}\||w_k^{t+1} - w^\*\||^2 = \mathbb{E}\||w_k^{t+1} - w_k^\*\||^2 + \||w_k^\* - w^\*\||^2
> $$
>
> The cross term vanishes by zero-mean noise. The first term contracts toward $w_k^*$ at rate $(1 - \eta\mu)^\tau$ under $L$-smoothness and $\mu$-strong convexity of $F_k$. The second term is the irreducible client-drift bias, bounded by $\zeta^2$. No interpolation is required.
>
> The $\alpha_{\min}$ factor enters only at aggregation (Step 4), where $\sum_k \alpha_k^t \cdot \mathbb{E}\|w_k^{t+1} - w^*\|^2$ gives the noise term $\eta^2\tau\sigma^2 / \alpha_{\min}$ via $\alpha_k^t \geq \alpha_{\min}$. Lemma 9 now presents the per-client bound without $\alpha_{\min}$.
>
> **C1b — The effective variance argument, Eq. (15).**
>
> The reviewer is correct that $\sigma^2_{\mathrm{eff}} \leq \sigma^2$ does not follow from PH stability alone. It requires the topology-gradient alignment condition:
>
> $$
> \||\phi_k - \phi_j\||_2 \leq \delta \implies \||\nabla F_k(w) - \nabla F_j(w)\|| \leq \rho(\delta)
> $$
>
> In the revision, Theorem 7 states convergence unconditionally (FedAvg rate, error floor $\zeta^2$). The claim $\sigma^2_{\mathrm{eff}} < \sigma^2$ is a corollary conditional on Proposition 8's alignment assumption. We label this empirical, supported by performance gains, not derived from stability. We add a paragraph in §3.7 clarifying which aspects are structural (aggregation form, convergence rate) and which depend on the data-dependent alignment assumption (reduced error floor).
>
> ---
>
> **[2] Privacy analysis — Theorem 6**
>
> **C2a — Analysis of $\phi_k$ alone does not cover $(\phi_k, \theta_k)$.**
>
> The reviewer is exactly right. Theorem 6 answers a narrow question: does adding $\phi_k$ increase information beyond $\theta_k$ alone? The descriptor's sensitivity $\Delta_\phi \leq c_\delta$ is bounded independently of $n_k$, while gradient sensitivity $\Delta_G = 2B/n_k$ grows as the dataset shrinks. The descriptor presents a structurally narrower channel — but this says nothing about $\theta_k$. The revised §3.6 states this explicitly: pTopoFL inherits all standard FL privacy risks from $\theta_k$. Theorem 6 quantifies only the marginal contribution of $\phi_k$.
>
> **C2b — No formal $(\varepsilon, \delta)$-DP guarantee.**
>
> The revision removes "privacy-preserving" from title, abstract, and all claims. The contribution is reframed as a narrower reconstruction surface — a structural property, not a formal DP guarantee. Section 3.6 opens with: "Theorem 6 is not a $(\varepsilon,\delta)$-DP guarantee." A formal DP composition would require DP-SGD on $\theta_k$ and a Gaussian mechanism on $\phi_k$ with sensitivity $\Delta_\phi \leq c_\delta$. Since $c_\delta < 2B/n_k$, the descriptor requires less noise. Deriving the tight bound is left to future work.
>
> We believe these changes fully address the reviewer's concerns while strengthening the paper's precision and its limitations.

---

### Review · Reviewer_V1u7 · 2026-07-19

**Summary Of Contributions:**

This paper proposes to modify the client and server operations in FL as follows: (i) clients exchange topological descriptors derived from persistent homology (PH), and (ii) the server first clusters the clients based on the Wasserstein similarity of their PH descriptors, then weighs the intra-cluster models by their topology, and finally aggregates the clusters with the global consensus. These changes allow for two simultaneous advantages enabled by the PH descriptors: (i) prevents data-reconstruction attacks by not sharing raw gradients, (ii) improves aggregation quality under non-IID client data distributions.

**Audience:**

Yes

**Audience Explanation:**

While I have some concerns regarding this paper, the ideas of using geometrical structures such as persistent homology for aggregation refinement in FL could be further studied. Also, the attempt to replace differential privacy with methods inherently resistant for data-reconstruction is worth pursuing.

**Broader Impact Concerns:**

The paper includes a satisfying discussion on broader impact concerns.

**Claims And Evidence:**

No

**Claims Explanation:**

1. It is hard to follow the Introduction and Background in Secs. 1 and 2.2 to understand how the authors arrived at using PH instead of gradient exchange. Current related work is lacking research on sharing other non-gradient parameters, which PH would be only one of them, to follow the progression from sharing raw gradients to the presented complex PH descriptors.

2. In the Introduction of Sec. 1, the authors first discuss proximal penalties, control variates and Moreau envelope as techniques to tackle client drift. While all of these approaches in FL have centralized counterparts discussed within those references, the paper does not discuss use cases of PH descriptors in centralized optimization, if there exists any. This lack of discussion makes it hard to appreciate why PH descriptors as one specific form geometric structures are proving to be useful in this paper. A few papers are presented later in Sec. 6 in Related Work, however, it seems those papers are using PH descriptors as a layer for feature engineering, not to replace gradient-based operations.

3. In the Introduction of Sec. 1, it is stated that "persistence diagrams form a metric space under the p-Wasserstein distance $W_p$". Is this a well-known result (please add references if so), or is it a result proved by this paper?

4. In Sec. 1 when the authors are presenting the five interconnected components, it seems the PH descriptors are only used to determine the aggregation weights, but $w^{t+1} = \sum_{k=1}^K{\alpha_k w_k^{t+1}}$ seems to still indicate that the clients are exchanging their model parameters. Furthermore, if PH descriptors are shared by the clients to the server, what does the server share back with the clients?

5. In Sec. 3.1, it is again unclear why the Betti numbers $\beta_j^{(k)}$ or the Betti curve $\\{ b_{\ell}^j \\}_{\ell=1}^L$ are useful (necessary and/or sufficient) geometrical descriptors, which are informative enough to replace the gradient exchanges (alongside other descriptors given in Eq. (5)). Lack of references makes it harder to appreciate the usefulness of these descriptors.

6. In Sec. 3.1, are Assumptions (A3)-(A5) also standard in the literature? Having some references which have made similar assumptions will greatly help. Also, recent papers in FL have been providing results for non-convex loss functions as well, not requiring Assumption (A2) on strong convexity.

7. One of the motivations of this paper is to improve the privacy of FL communications, without the need to use differential privacy (DP). While it is understandable that a $(\epsilon, \delta)$-type analysis might not be possible for this approach based on PH descriptors, it is still expected to make a comparison with FL+DP in numerical experiments of Sec. 4.

8. In Sec. 4.2, some of the results lack insight as the plots remain flat, e.g., in Figs. 2-(A), 2-(B), 4-(B), 5-(A) and 7.

**Requested Changes:**

Please see my comments on why some of the claims made in the submission are not supported by accurate, convincing and clear evidence.